# Exploring Real-Time Super-Resolution: Benchmarking and Fine-Tuning for Streaming Content

**Evgeney Bogatyrev[1,2,3], Khaled Abud[1,2,3], Ivan Molodetskikh[1,2], Nikita Alutis[1,2],
Dmitriy Vatolin[3]**
[1]AI Center, Lomonosov Moscow State University
[2]Lomonosov Moscow State University, 119991, Moscow, Russia
[3]MSU Institute for Artificial Intelligence, Lomonosov Moscow State University
```
evgeney.bogatyrev@graphics.cs.msu.ru
khaled.abud@graphics.cs.msu.ru
ivan.molodetskikh@graphics.cs.msu.ru
nikita.alutis@graphics.cs.msu.ru
dmitriy@graphics.cs.msu.ru
```

## Abstract

Recent advancements in real-time super-resolution have enabled higher-quality video streaming, yet existing methods struggle with the unique challenges of compressed video content. Commonly used datasets do not accurately reflect the characteristics of streaming media, limiting the relevance of current benchmarks. To address this gap, we introduce a comprehensive dataset - **StreamSR** - sourced from YouTube, covering a wide range of video genres and resolutions representative of real-world streaming scenarios. We benchmark 11 state-of-the-art real-time super-resolution models to evaluate their performance for the streaming use-case.

Furthermore, we propose **EfRLFN**, an efficient real-time model that integrates Efficient Channel Attention and a hyperbolic tangent activation function - a novel design choice in the context of real-time super-resolution. We extensively optimized the architecture to maximize efficiency and designed a composite loss function that improves training convergence. EfRLFN combines the strengths of existing architectures while improving both visual quality and runtime performance.

Finally, we show that fine-tuning other models on our dataset results in significant performance gains that generalize well across various standard benchmarks. We made the dataset, the code, and the benchmark available at `https://github.com/EvgeneyBogatyrev/EfRLFN`.

## 1 Introduction

In recent years, the demand for high-quality video streaming has surged, driven by the increasing adoption of platforms such as YouTube, Twitch, and Netflix (Grand View Research, 2025). To balance user experience with bandwidth constraints, streaming services often compress video content, which may introduce artifacts such as blockiness, blurring, and loss of fine details. While real-time super-resolution (SR) methods have emerged as a solution to enhance video quality, existing techniques struggle to address the unique challenges posed by heavily compressed videos in real-world scenarios (Bogatyrev et al., 2024).

Further investigations suggest that there are a limited number of SR methodologies capable of effectively upscaling compressed videos (Li et al., 2021b). Additionally, there is a notable scarcity of models designed for real-time video upscaling. The NTIRE challenges (Conde et al., 2023; Ren et al., 2024) have played a pivotal role in advancing the development of real-time SR models, fostering significant progress in this domain. However, it is worth mentioning that many of these SR models have been primarily trained on datasets such as DIV2K (Agustsson & Timofte, 2017) or Vimeo90K (Xue et al., 2019), which do not adequately represent compressed video scenarios.

NVIDIA's Video Super-Resolution (NVIDIA VSR), a learning-based SR method integrated into GPU driver, demonstrates the growing focus on real-time SR for streaming applications (Choi, 2023). However, despite its efficiency, NVIDIA VSR often fails to restore fine textures and introduces over-smoothing, limiting its effectiveness, as illustrated in Figure 1. Similarly, other state-of-the-art SR methods, such as SPAN (Wan et al., 2024) and RLFN (Kong et al., 2022), demonstrate promising results on standard datasets but are not optimized for the complexities of streaming content (Bogatyrev et al., 2024).

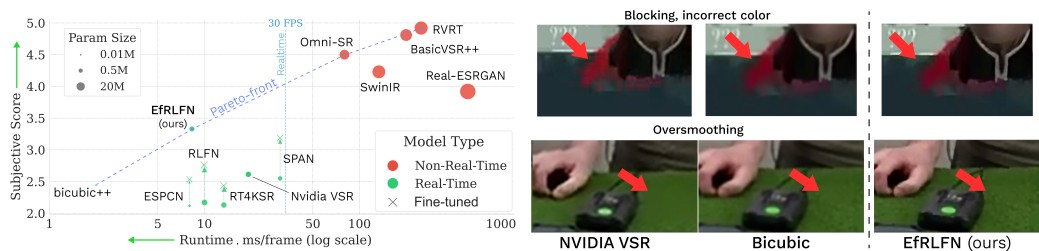

Figure 1: **Left**: Trade-off between user preference score and runtime speed for various 2× super-resolution models. Blue line represents Pareto-optimal front. Models achieving real-time performance[2] are shown in green, while slower models are red. "×" represent models fine-tuned on StreamSR dataset. **Right**: Examples of NVIDIA VSR artifacts compared to the proposed EfRLFN model and bicubic interpolation.

To address these challenges, we present **StreamSR**: a comprehensive dataset of 5,200 YouTube videos (and more than 10M individual frames) covering diverse genres and resolutions, offering a robust benchmark for evaluating SR methods under streaming conditions. Additionally, we propose **Efficient Residual Lightweight Feature Network (EfRLFN)**. This model is specifically optimized for real-time applications, combining techniques adopted from prior work in real-time SR and related fields. EfRLFN achieves significant improvements in terms of both objective and subjective video quality metrics while maintaining computational efficiency.

Despite the promising results of video-based models on compressed video tasks, their architectures are usually too complex to meet the FPS requirements for real-time deployment. For this reason, the proposed EfRLFN model was intentionally developed as an image SR model, with the goal of combining effective methodologies and optimizing existing architectures to achieve superior performance.

Our **main contributions** are as follows:

1. **StreamSR – Streaming Super-Resolution Dataset**. We collect and analyze a multiscale dataset derived from YouTube, comprising 5,200 carefully filtered user-generated videos of diverse topics and content types, and conduct a comparative analysis with existing datasets.

2. **EfRLFN – Efficient Real-Time Super-Resolution Method**. Building upon a popular RLFN architecture, we thoroughly optimize it with a new block design, Efficient Channel Attention and enhanced training procedure. Our evaluations show that EfRLFN establishes a new SoTA in real-time SR, outperforming other methods in quality-complexity tradeoff, supported by extensive ablation experiments.

3. **Systematic benchmarking of real-time SR models**. We evaluate 11 real-time super-resolution methods using our dataset, assessing their performance both with and without fine-tuning on the training set. We employ 7 different objective image quality metrics and conduct a large-scale user preference study with more than 3,800 participants. User preference dataset will be also published alongside the SR benchmark.

---

[2]≥30 FPS on a Nvidia RTX 2080 GPU, which approximates a potential user setup

## 2 RELATED WORK

### 2.1 SUPER-RESOLUTION METHODS

Recent advances in deep learning have significantly improved super-resolution (SR) techniques, with particular progress in real-time applications. Current approaches achieve varying balances between reconstruction quality and computational efficiency through innovative architectural designs.

Among real-time capable models, **Bicubic++** (Bilecen & Ayazoglu, 2023) combines fast reversible feature extraction with global structured pruning of convolutional layers, achieving processing speeds comparable to traditional bicubic interpolation. **AsConvSR** (Guo et al., 2023) introduces dynamic assembled convolutions that adapt their kernels based on input characteristics, optimized particularly for mobile and edge device deployment. For high-resolution upscaling, **RT4KSR** (Zamfir et al., 2023) employs pixel-unshuffling and structural re-parameterization of NAFNet blocks to efficiently handle 720p/1080p to 4K conversion. The **SPAN** architecture (Wan et al., 2024) features a parameter-free attention mechanism with symmetric activations to amplify important features and suppress redundant ones, reducing computational overhead by 40% compared to conventional attention modules. Similarly, **RLFN** (Kong et al., 2022) implements a streamlined three-layer convolutional structure within a residual learning framework, achieving 1080p to 4K conversion in under 15ms while maintaining strong feature aggregation. **TMP** (Zhang et al., 2024) enhances online video super-resolution by minimizing computational redundancy in motion estimation.

In addition to lightweight real-time methods, transformer-based models like **SwinIR** (Liang et al., 2021) and **HiT-SR** (Zhang et al., 2025) achieve superior quality through self-attention mechanisms and hierarchical multi-scale processing. There are other approaches to SR task, such as **COMISR** (Li et al., 2021b), which specialize in compressed video upscaling. **Real-ESRGAN** (Wang et al., 2021) is a popular GAN-based method that employs high-order degradation modeling to simulate real-world degradation. Although these methods demonstrate superior results compared to real-time SR models, they are typically computationally intensive, making them unsuitable for real-time applications.

A growing line of research specifically targets compression-aware video super-resolution, where models are designed to handle distortions introduced by modern video codecs. Representative works include **CAVSR** (Wang et al., 2023b) and **TAVSR** (Wei et al., 2024), which leverage codec-related features to restore severely degraded compressed sequences. Recent approaches such as **FTVSR** (Qiu et al., 2022) and **FTVSR++** (Qiu et al., 2023) utilize transformer architectures that perform self-attention across joint space–time–frequency domains. Other works (An et al., 2025) incorporate diffusion-based denoising and distortion-control modules to reduce the impact of compression artifacts during generation. These approaches typically achieve strong reconstruction quality on low-bitrate inputs but rely on computationally heavy architectures, making them unsuitable for real-time deployment.

While all these models have demonstrated varying degrees of success in enhancing image quality through super-resolution techniques, challenges remain in effectively addressing the issues posed by compressed video content. In this paper, we build upon this foundational works aiming to evaluate and incorporate best practices to create the most suited SR model for real-time use case.

### 2.2 DATASETS AND REAL-TIME SR CHALLENGES

Datasets play a crucial role in training SR models. While **DIV2K** (Agustsson & Timofte, 2017) and **Vimeo90K** (Xue et al., 2019) provide clean HR-LR pairs, they lack streaming compression artifacts. The **YouTube-8M** dataset (Abu-El-Haija et al., 2016) offers diverse content but not multi-scale real-time scenarios. Additionally, several benchmark datasets are widely used for evaluating SR performance, including Set14 (Zeyde et al., 2012), BSD100 (Silva, 2001), and Urban100 (Huang et al., 2015), which contain natural and structured images at fixed scales. For video super-resolution tasks, REDS (Nah et al., 2019) provides dynamic scenes with motion blur and compression artifacts, making it suitable for training and testing temporal SR models.

Recent SR advancements focus on real-time efficiency. The **NTIRE 2023 Real-Time SR Challenge** (Conde et al., 2023) promoted lightweight architectures using pruning, quantization, and knowledge distillation for live applications. Its successor, **NTIRE 2024 Efficient SR challenge** (Ren et al., 2024), emphasized resource efficiency with adaptive transformers and multi-task learning

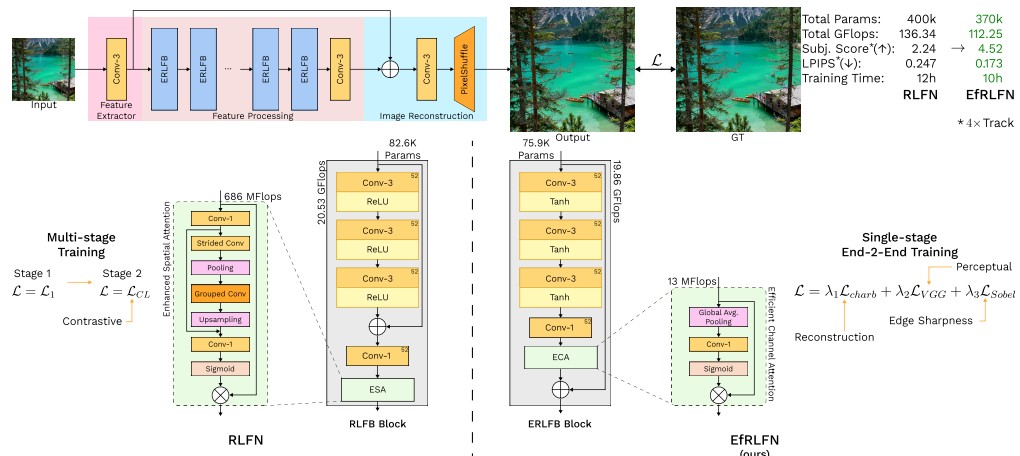

Figure 2: Visual summary of the proposed EfRLFN model and the comparison with the original RLFN architecture.

combining SR with denoising and deblocking. The latest **NTIRE 2025 challenge** (Ren et al., 2025) further advanced efficiency by balancing runtime, FLOPs, and parameters, introducing innovations like state-space models and ConvLora-based distillation.

While these challenges rank real-time SR methods, their exclusion of compressed video limits their applicability to streaming content. With this idea in mind, we created a novel dataset which includes diverse videos with compression at several resolutions.

## 3 PROPOSED MODEL

### 3.1 NETWORK ARCHITECTURE

Although video SR models excel in standard SR tasks, their high complexity hinders real-time processing and batch training. Thus, we focus on image SR for our approach. The proposed **Efficient Residual Local Feature Network (EfRLFN)** enhances the RLFN (Kong et al., 2022) architecture through targeted modifications to improve efficiency and reconstruction quality. Figure 2 demonstrates the overall architecture of the model and its core modifications over RLFN. EfRLFN consists of a feature extraction block, main feature processing module, and an output reconstruction block. Due to their simplicity and parameter efficiency, we retain the feature extractor and reconstruction modules from the original RLFN model. The key innovations of our model are focused on ERLFB block design and the training procedure.

**Efficient Residual Local Feature Blocks (ERLFB)** replace the original RLFB blocks in RLFN. The primary change involves adopting **tanh activation** in the refinement modules instead of ReLU. This modification is motivated by recent findings (Wan et al., 2024) demonstrating that odd symmetric activation functions, such as $Sigmoid(x) - 0.5$ or tanh, preserve both the magnitude and sign of features. Unlike ReLU, which discards negative activations, tanh ensures richer gradient flow and prevents directional information loss in attention maps, leading to more accurate feature refinement. Empirical studies demonstrate that tanh-based activation can enhance gradient propagation in deep networks (Clevert et al., 2015) and improve performance in dense prediction tasks, including image-to-image translation (Isola et al., 2017; Wang et al., 2018).

Another important modification lies in the attention mechanism used for global information aggregation. **Efficient Channel Attention (ECA)** substitutes the Enhanced Spatial Attention (ESA) mechanism used in RLFN. While ESA employs multi-layer convolutional groups for spatial attention, ECA leverages a lightweight channel attention mechanism based on global average pooling and a $1 \times 1$ convolution. This change significantly reduces computational overhead while maintaining competitive performance, as validated in our ablation studies.

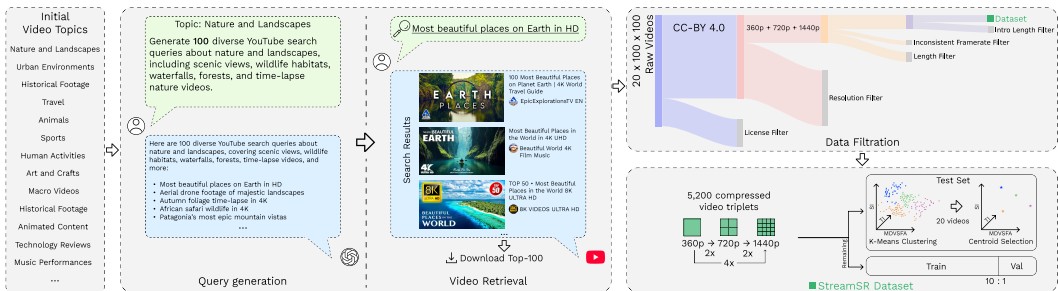

Figure 3: The process of video collection for our StreamSR dataset. The resulting set is split into train, test, and validation parts. **Zoom in for better clarity.**

Table 1: Comparison of UGC Video Datasets. Quality range shows 5th-95th percentile scores of MDTVSFA metric. Duration shows the typical duration of a video clip. *Image* means that dataset contains only single images.

| Dataset | Year | Size | Resolutions | LR? | Quality Range | Use-Case | Duration |
|---|---|---|---|---|---|---|---|
| YouTube-8M (Abu-El-Haija et al., 2016) | 2016 | 8M videos | 240p–1080p | ✗ | 0.45–0.58 | Classification | 3–5 min |
| Kinetics-400 (Kay et al., 2017) | 2017 | 240k clips | 1080p | ✗ | 0.48–0.59 | Action Recognition | 8–12 sec |
| Kinetics-700 (Carreira et al., 2019) | 2019 | 650k clips | 1080p | ✗ | 0.47–0.58 | Action Recognition | 8–12 sec |
| YouTube-VOS (Xu et al., 2018) | 2018 | 4.5k videos | 360p–1080p | ✗ | 0.46–0.57 | Object Segmentation | 15–25 sec |
| YouTube-BB (Real et al., 2017) | 2017 | 380k videos | 1080p | ✗ | 0.50–0.60 | Object Tracking | 8–12 sec |
| YT-Temporal 180M (Zellers et al., 2021) | 2020 | 180M clips | 144p–1080p | ✗ | 0.42–0.56 | Language Pretraining | 3–7 sec |
| DIV2K (Agustsson & Timofte, 2017) | 2017 | 1,000 images | 2K | ✓ | 0.55–0.65 | Image SR | *Image* |
| REDS (Nah et al., 2019) | 2019 | 300 clips | 360p–720p | ✓ | 0.50–0.60 | Video SR | 2–5 sec |
| Vimeo-90K (Xue et al., 2019) | 2019 | 90K clips | 448p | ✗ | 0.48–0.58 | Video SR | 1 sec |
| UDM10 (Yi et al., 2019) | 2019 | 10K pairs | 540p–4K | ✓ | 0.45–0.58 | Image SR | *Image* |
| RealSR (Cai et al., 2019) | 2019 | 595 pairs | 4K | ✓ | 0.43-0.59 | Real-world SR | *Image* |
| VideoLQ (Chan et al., 2022) | 2021 | 1,000 clips | 270p–1080p | ✓ | 0.43–0.54 | Video SR | 5–10 sec |
| **StreamSR (Proposed)** | 2025 | 5.2k videos | 360p–1440p | ✓ | **0.41–0.61** | Super-Resolution | 25–30 sec |

Additionaly, the **reconstruction path** of the ERLFB is streamlined by removing redundant skip connections and simplifying the feature smoothing step to a single $3 \times 3$ convolution. This optimization reduces computational fragmentation, improving inference speed without compromising reconstruction quality.

## 3.2 LOSS FUNCTION

While RLFN's contrastive loss effectively aligns intermediate features through positive and negative sample pairs, it introduces two practical challenges: (1) sensitivity to the choice of feature extractor layers (shallow layers are required for PSNR-oriented tasks (Kong et al., 2022)), and (2) increased computational overhead due to pairwise feature comparisons. EfRLFN introduces a composite loss function designed to address the limitations of RLFN's contrastive loss while emphasizing structural fidelity and edge preservation. The loss consists of three key components:

$$\mathcal{L} = \underbrace{\lambda_{Charb}\mathcal{L}_{Charb}}_{\text{Reconstruction}} + \underbrace{\lambda_{VGG}\mathcal{L}_{VGG}}_{\text{Perception}} + \underbrace{\lambda_{Sobel}\mathcal{L}_{Sobel}}_{\text{Edge Sharpness}} \qquad (1)$$

**Charbonnier Loss** enforces pixel-level accuracy while providing robustness to outliers, addressing a limitation of the L1/L2 losses used in RLFN's early training stages. Its square-root formulation ensures stable gradients even for large residuals, making it particularly suitable for super-resolution tasks.

$$\mathcal{L}_{Charb} = \sqrt{\|\mathbf{I}_{HR} - \mathbf{I}_{SR}\|^2 + \epsilon^2}, \quad \epsilon > 0 \qquad (2)$$

where $I_{\text{HR}}$ denotes the reference high-resolution image and $I_{\text{SR}}$ denotes the output of the SR module.

**VGG Perceptual Loss** leverages deep features from a pre-trained VGG-19 network (Simonyan & Zisserman, 2014) to guide the model toward perceptually realistic outputs.

$$\mathcal{L}_{VGG} = \|\phi_{\text{VGG}}(\mathbf{I}_{HR}) - \phi_{\text{VGG}}(\mathbf{I}_{SR})\|_1, \tag{3}$$

where $\phi_{\text{VGG}}(\cdot)$ denotes ReLU activations from the *conv5_4* layer. While RLFN employs a contrastive loss to align intermediate features, our VGG loss provides analogous perceptual supervision without requiring complex feature pairing or negative sampling. This simplification reduces training complexity while maintaining high visual quality.

**Sobel Edge Loss** explicitly optimizes edge sharpness by penalizing discrepancies in gradient maps between the super-resolved and ground truth images.

$$\mathcal{L}_{Sobel} = \|S(\mathbf{I}_{HR}) - S(\mathbf{I}_{SR})\|_2^2, \tag{4}$$

where $S(\cdot)$ applies the Sobel operator to extract horizontal and vertical edges.

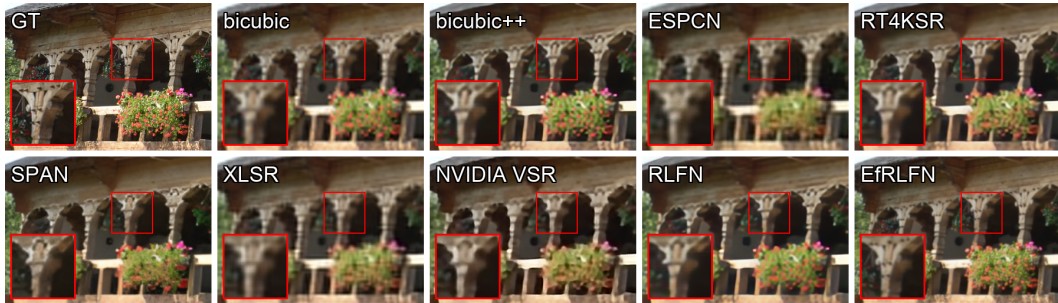

Figure 4: The comparison between EfRLFN and several $4\times$ real-time SR models.

This component addresses a critical gap in RLFN's approach, which relies on implicit edge preservation through contrastive learning on shallow features. By directly targeting edge quality, our loss ensures crisper and more confident reconstructions, particularly for high-frequency details. User studies confirm that such edge-enhanced outputs align with human perceptual preferences, achieving significantly higher subjective ratings (Zhang et al., 2018; Bogatyrev et al., 2024).

Overall, EfRLFN outperforms RLFN with 15% faster inference via hardware-friendly ECA and tanh activations, faster and more stable training through a simplified single-stage training with a carefully designed composite loss, and better quality as measured with objective and subjective metrics. We further discuss the impact of individual design choices in the Ablation Study.

## 4    REAL-TIME SR BENCHMARK

In this section, we detail the methodology employed for video collection process for the StreamSR dataset (summarized in Figure 3), their subsequent categorization, and overall evaluation protocol and benchmark setup.

### 4.1    DATASET COLLECTION

**Video Retrieval**   We employed a Large Language Model (GPT-4o) (Achiam et al., 2023) to generate search queries for collecting videos. Starting with 20 distinct YouTube categories (e.g., travel, education, gaming), the LLM produced 100 diverse YouTube search queries per category. We then downloaded the top 100 videos for each query. The list of initial topics, LLM prompts and examples of queries are provided in the Section 17).

**Filtering**   Only Creative Commons (*CC BY 4.0*) videos were included, with available resolutions of 360p, 720p, and 1440p. Videos had to maintain a consistent frame rate, verified automatically, to enable precise frame alignment. Resolutions were selected to support two SR benchmark tracks: $2\times$

Table 2: Comparison of real-time SR methods with 95% confidence intervals on $2\times$ super-resolution track. Best results are in **bold**, second best are underlined. "$T$" indicates fine-tuning on StreamSR. "Subj." represents the Bradley-Terry subjective scores. Extended results are in Section 19.

| Method | Our Benchmark | | | | Standard Benchmarks | | | FPS↑ |
|---|---|---|---|---|---|---|---|---|
| | Subj.↑ | PSNR↑ | LPIPS↓ | CLIP-IQA↑ | BSD100 SSIM↑/LPIPS↓ | Urban100 SSIM↑/LPIPS↓ | DIV2K SSIM↑/LPIPS↓ | |
| AsConvSR$^T$ | 1.93±0.11 | 35.25±0.15 | 0.214±0.008 | 0.48±0.02 | 0.832 / 0.270 | 0.824 / 0.208 | 0.883 / 0.206 | 213±8 |
| RT4KSR | 2.40±0.12 | 36.45±0.16 | 0.213±0.009 | 0.49±0.02 | 0.805 / 0.296 | 0.769 / 0.244 | 0.860 / 0.221 | 102±5 |
| RT4KSR$^T$ | 2.43±0.11 | 37.55±0.14 | 0.070±0.003 | 0.53±0.02 | 0.812 / 0.254 | 0.766 / 0.223 | 0.864 / 0.186 | 102±5 |
| bicubic | 2.44±0.14 | 30.32±0.21 | 0.076±0.005 | 0.44±0.03 | 0.752 / 0.386 | 0.711 / 0.324 | 0.820 / 0.292 | **1829±50** |
| ESPCN | 2.48±0.12 | 30.71±0.18 | 0.078±0.004 | 0.45±0.02 | 0.774 / 0.534 | 0.505 / 0.437 | 0.514 / 0.515 | 201±7 |
| ESPCN$^T$ | 2.49±0.11 | 35.76±0.15 | 0.072±0.003 | 0.51±0.02 | 0.814 / 0.236 | 0.840 / 0.150 | 0.862 / 0.180 | 201±7 |
| Bicubic++$^T$ | 2.44±0.13 | 36.79±0.15 | 0.087±0.004 | 0.52±0.02 | 0.768 / 0.360 | 0.725 / 0.298 | 0.832 / 0.270 | 1629±45 |
| NVIDIA VSR | 2.57±0.14 | 37.40±0.13 | 0.082±0.003 | 0.56±0.02 | 0.788 / 0.243 | 0.786 / 0.152 | 0.858 / 0.180 | 52±3 |
| XLSR$^T$ | 2.56±0.12 | 37.25±0.14 | 0.230±0.010 | 0.47±0.02 | 0.817 / 0.244 | 0.828 / 0.158 | 0.864 / 0.183 | 429±15 |
| SMFANet$^T$ | 2.37±0.14 | 37.14±0.15 | 0.158±0.007 | 0.51±0.02 | 0.803 / 0.239 | 0.798 / 0.153 | 0.865 / 0.177 | 327±12 |
| SAFMN$^T$ | 2.26±0.15 | 37.09±0.15 | 0.122±0.006 | 0.53±0.02 | 0.813 / 0.236 | 0.808 / 0.150 | 0.871 / 0.175 | 273±10 |
| RLFN | 2.17±0.15 | 37.03±0.15 | 0.086±0.004 | 0.54±0.02 | 0.805 / 0.238 | 0.803 / 0.147 | 0.876 / 0.175 | 225±8 |
| RLFN$^T$ | 2.69±0.13 | 37.63±0.12 | 0.072±0.003 | 0.58±0.02 | 0.834 / 0.239 | 0.845 / 0.153 | 0.881 / 0.178 | 225±8 |
| SPAN | 2.55±0.12 | 37.45±0.13 | 0.066±0.003 | 0.57±0.02 | 0.836 / 0.222 | 0.841 / 0.139 | 0.887 / 0.168 | 60±3 |
| SPAN$^T$ | 3.13±0.15 | 37.73±0.12 | 0.063±0.003 | 0.61±0.02 | 0.837 / 0.239 | 0.847 / 0.118 | 0.890 / 0.175 | 60±3 |
| **EfRLFN$^T$** | **3.33±0.14** | **37.85±0.11** | **0.059±0.003** | **0.65±0.02** | **0.847 / 0.190** | **0.856 / 0.116** | **0.892 / 0.145** | 271±10 |
| *Non-real-time SR models* | | | | | | | | |
| Real-ESRGAN | 3.87±0.13 | 37.65±0.12 | 0.048±0.007 | 0.66±0.02 | 0.857 / 0.172 | 0.867 / 0.112 | 0.901 / 0.137 | 9±1 |
| BasicVSR++ | 4.87±0.13 | 38.05±0.13 | 0.037±0.007 | 0.70±0.02 | 0.860 / 0.158 | 0.868 / 0.110 | 0.907 / 0.131 | 15±1 |
| COMISR | 4.32±0.14 | 38.15±0.12 | 0.033±0.006 | 0.67±0.03 | 0.894 / 0.161 | 0.871 / 0.108 | 0.913 / 0.125 | 7±1 |

(720p→1440p) and $4\times$ (360p→1440p). For each resolution, the highest-bitrate stream was chosen to ensure optimal quality of the target high-res stream. We extracted the first 30 seconds of each video ($\leq$2,000 frames). Videos with long static intros were discarded with a scene-difference filter analyzing the $1^{st}$, $100^{th}$ and $150^{th}$ frames. This yielded a collection of 5,200 videos, which we named **StreamSR** dataset.

**Categorization** To construct a diverse and representative test set, we clustered videos using SI, TI (Wang et al., 2019), bitrate, MDTVSFA (Li et al., 2021a), and SigLIP (Zhai et al., 2023) text embeddings of the search queries. SigLIP embeddings were projected into 3-dimentional space with PCA. From each of the 20 resulting clusters, the video closest to the centroid was selected. The rest were split into training and validation sets in a 10:1 ratio.

Table 1 compares our StreamSR dataset with existing UGC datasets. Unlike classification (YouTube-8M, Kinetics) or segmentation datasets (YouTube-VOS), StreamSR is specifically designed for super-resolution, providing aligned 360p-1440p videos with 2×/4× upscaling tracks. With longer 25-30s clips, LR-HR pairs with natural compression degradations, and broader quality range, it better represents real-world streaming scenarios than previous SR datasets (REDS, Vimeo-90K).

## 4.2 BENCHMARK SETUP

For benchmarking and fine-tuning, we selected a comprehensive list of SR models based on the year-to-year results of the NTIRE challenges, ranging from classical approaches like bicubic, bicubic++ (Bilecen & Ayazoglu, 2023) and ESPCN (Talab et al., 2019) up to the recent SoTA methods like SMFANet (Zheng et al., 2024), RLFN (Kong et al., 2022) and SPAN (Wan et al., 2024). We had also included a proprietary NVIDIA VSR model to the comparison. Table 5 in the Appendix summarizes the information about the tested models.

We selected a diverse set of objective metrics that assess different aspects of SR algorithms' performance. PSNR and SSIM (Wang et al., 2004) serve as fundamental measures of pixel-level fidelity and structural similarity. LPIPS (Zhang et al., 2018), on the other hand, goes beyond traditional metrics, and employs deep features to better align with human perception. Aside from Full-Reference metrics, we also included No-Reference (NR) models in the comparison. They assess the quality of super-resolved images independently, without comparing them to the real high-resolution examples. Specifically, we employed 2 NR metrics: MUSIQ (Ke et al., 2021), a powerful transformer-based model that excels in real-world content assessment, and CLIP-IQA (Wang et al., 2023a), a popular model that leverages vision-language correspondence to capture semantic preservation in SR outputs.

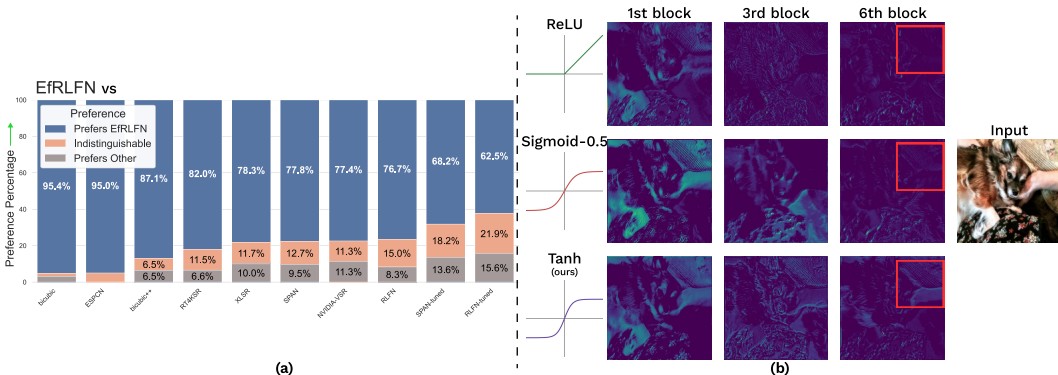

Figure 5: (a) Pairwise preference evaluation of EfRLFN against other real-time super-resolution methods. (b) A comparison of the output feature maps from the first, third, and sixth ERLFB blocks. The features are taken from the output of the ECA block within each ERLFB.

We evaluate the alignment of these metrics with human preferences in Section 15, proving their applicability to the super-resolution quality assessment task.

The subjective comparison still remains the best option for evaluating SR performance. Since user studies can often be expensive, we preliminary excluded models that demonstrated subpar performance based on objective metrics. To evaluate and rank the performance of various SR models from a subjective perspective, we conducted a pair-wise crowd-sourced comparison utilizing the Subjectify.us (Subjectify.us, 2025) service. Comparison included 11 SR models (+ Ground Truth) and 660 video pairs in total.

A total of 3,822 unique individuals participated in our subjective evaluation. Consequently, the final subjective scores were derived from the 37,184 valid responses, applying the Bradley-Terry model (Bradley & Terry, 1952). More detailed information on this study is provided in Section 14.

## 5 EXPERIMENTS

**Fine-tuning** To ensure a fair benchmarking of each model, we pre-trained all tested models on the training set of our dataset and evaluated the results of both pre-trained and non-pre-trained checkpoints. If a model did not have pre-trained weights for a specific upscale factor ($2\times$ or $4\times$), we trained it from scratch and utilized only the trained version. Additional information on the training can be found in Section 12.

**Validation on the StreamSR** Table 2 (left) shows that the proposed model, EfRLFN, achieves superior performance on the test set of our dataset among the real-time models. The performance increase generalizes well across various objective metrics, and is further confirmed by the user study (Figure 5(a)). The results also indicate that SPAN, RLFN, and ESPCN models benefit significantly from fine-tuning on our dataset: there is a notable positive performance gap in both objective and subjective evaluations. After tuning, some of these models outperform the proprietary NVIDIA VSR model in user preferences (Figure 1). We also include some non-real-time SR models for comparison.

Figure 5(a) demonstrates the results of pair-wise subjective comparison with other SR models: evidently, users largely prefer the outputs of EfRLFN over other methods. Specifically, EfRLFN is favored over NVIDIA VSR in 77.4% of cases. The only competitive alternatives (SPAN-tuned and RLFN-tuned) are also trained on the StreamSR training set.

**Validation on Other Datasets** We validated EfRLFN and other real-time SR methods on five widely recognized datasets: Set14 (Zeyde et al., 2012), BSD100 (Silva, 2001), Urban100 (Huang et al., 2015), REDS (Nah et al., 2019), and the validation set of DIV2K (Agustsson & Timofte, 2017). The average metric results are presented in Table 2 (right). EfRLFN consistently outperforms all competing approaches across all evaluated scenarios, achieving superior performance in both $2\times$

and $4\times$ resolution upscaling. Other quality assessment metrics, as well as the $4\times$ dataset tracks are reported in Section 19.

Figures 4 and 6(b) provide a visual comparison between different models, further demonstrating the effectiveness of EfRLFN in real-time super-resolution. Visual comparisons clearly demonstrate EfRLFN's advantages: it produces sharper images with more accurate details, while competing models often struggle to reproduce clear boundaries and fine textures, resulting in noticeable artifacts or blurring in challenging regions.

Our experiments reveal that training on our proposed dataset not only benefits EfRLFN but also improves the performance of other competing models, as evidenced by the quantitative results in Table 2 (particularly noticeable in cases such as SPAN and RLFN).

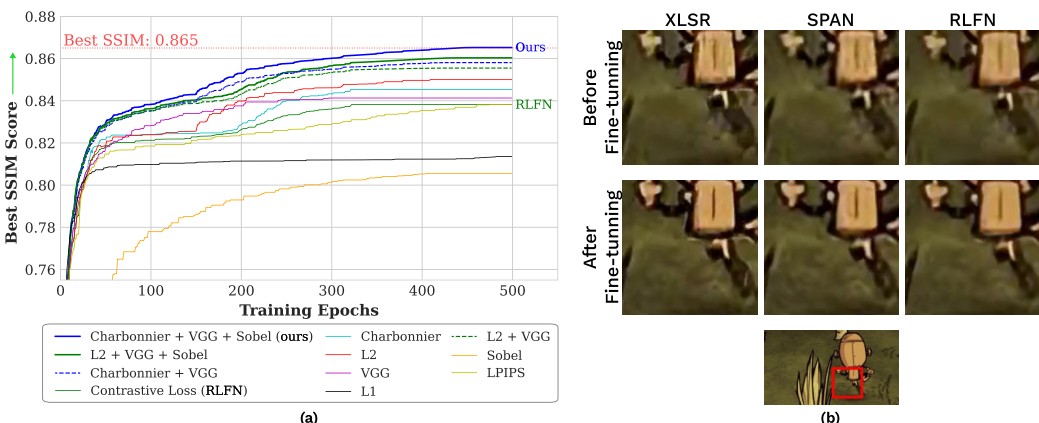

Figure 6: (a) The impart of loss functions on the training of EfRLFN on $4\times$ track of the proposed dataset. "Contrastive Loss" represents the original training procedure of RLFN model. (b) The comparison between SR models before and after fine-tuning on our StreamSR dataset.

## 5.1 ABLATION STUDY

**Loss Analysis** Figure 6(a) illustrates the impact of different loss functions on the SSIM performance of EfRLFN during training on the 4× track of our dataset. The full combination of Charbonnier, VGG, and Sobel losses (blue line) achieves the highest SSIM score of 0.865, clearly outperforming all ablations. This demonstrates the complementary strengths of each loss: Charbonnier contributes to structural accuracy, VGG to perceptual consistency, and Sobel to edge preservation. Ablating any component results in noticeable drops, particularly the removal of Charbonnier or VGG, which significantly impairs SSIM convergence. Simpler loss terms like $L_1$, $L_2$, and LPIPS perform substantially worse throughout training, highlighting the importance of combining both distortion-based and perceptual components for optimal SR quality. Notably, our approach outperforms RLFN in both quality and training efficiency. Unlike RLFN, which employs a two-stage training procedure with $L_1$ and contrastive losses, we train our network end-to-end, achieving superior visual quality while reducing training time by 16%.

**Architectural Analysis** Table 3 presents an ablation study isolating the effects of activation functions and attention mechanisms in EfRLFN. The combination of hyperbolic tangent activation and ECA attention yields the best overall performance, achieving the highest SSIM, strong perceptual quality, and the fastest runtime with a minimal parameter count. Replacing $tanh$ with a shifted sigmoid noticeably degrades SSIM and LPIPS across both attention variants. Similarly, using the heavier ESA attention block leads to lower frame rates and marginal gains in

Table 3: Ablation Study on Activation functions and Attention modules for EfRLFN. The best results appear in **bold**. $S(x)$ denotes a sigmoid function.

| Components | | Metrics | | | |
|---|---|---|---|---|---|
| **Activation** | **Attention** | **SSIM ↑** | **LPIPS ↓** | **FPS ↑** | **Params ↓** |
| $tanh$ | ECA | **0.865** | 0.173 | **314** | **0.37M** |
| $tanh$ | ESA | 0.863 | **0.171** | 234 | 0.4M |
| $S(x) - 0.5$ | ECA | 0.856 | 0.179 | 305 | 0.37M |
| $S(x) - 0.5$ | ESA | 0.852 | 0.181 | 237 | 0.4M |
| $ReLU$ | ECA | 0.847 | 0.184 | 303 | 0.37M |
| $ReLU$ | ESA | 0.849 | 0.182 | 235 | 0.4M |

LPIPS at best. These results indicate that $tanh$ activation paired with the lightweight ECA module provides an optimal balance between quality and efficiency for real-time super-resolution.

Figure 5(b) shows output features from each ERLFB block of a model trained with different activation functions. Note that non-odd activation functions (ReLU) result in poor feature quality. Compared to $Sigmoid - 0.5$, $tanh$ activation preserves more high-frequency features, which correlates with better detail restoration. This results in more detailed and cleaner outputs from the EfRLFN model.

## 5.2 ONNX RUNTIME RESULTS

To complement the PyTorch benchmarks, we exported EfRLFN, RLFN, and SPAN to ONNX and evaluated them using ONNX Runtime (v1.19.2) with both CUDA and TensorRT (v10.4.0) execution providers. All experiments were conducted on an NVIDIA RTX A6000 GPU using CUDA 11.7. The models were exported and executed in FP16 precision, and latency was measured on 360×480 inputs for both 2× and 4× upscaling.

As shown in Table 4, EfRLFN consistently achieves lower latency than RLFN across both execution providers and surpasses real-time throughput ($\geq$ 30 FPS). The model also benefits significantly from TensorRT optimizations, further demonstrating its suitability for deployment-oriented inference engines.

| Model | CUDA (ms) | TensorRT (ms) | Speedup | FPS (TRT) | Subjective Score |
|---|---|---|---|---|---|
| | | $2\times$ results | | | |
| RLFN | 30.76 | 29.12 | $1.06\times$ | 34.3 | $2.69 \pm 0.15$ |
| SPAN | 16.29 | 10.46 | $1.55\times$ | 95.6 | $2.55 \pm 0.12$ |
| EfRLFN (ours) | 17.43 | 12.07 | $1.44\times$ | 82.8 | $\mathbf{3.33 \pm 0.14}$ |
| | | $4\times$ results | | | |
| RLFN | 38.24 | 32.48 | $1.18\times$ | 30.8 | $4.32 \pm 0.13$ |
| SPAN | 17.20 | 10.86 | $1.58\times$ | 92.1 | $3.14 \pm 0.12$ |
| EfRLFN (ours) | 20.25 | 14.66 | $1.38\times$ | 68.2 | $\mathbf{4.52 \pm 0.13}$ |

Table 4: ONNX Runtime results for $2\times$ and $4\times$ upscaling. FPS is measured with TensorRT.

## 6 CONCLUSION

Our research advances real-time super-resolution by introducing a comprehensive multi-scale dataset of 5,200 compressed videos, specifically designed for streaming applications. This dataset, coupled with systematic benchmarking of 11 SR models, provides valuable performance insights and establishes a robust evaluation framework. All resources - including the benchmark, dataset, and raw subjective comparison data (37,184 responses) - are publicly available at `https://github.com/EvgeneyBogatyrev/EfRLFN`. These subjective evaluations can serve as valuable data for training video quality assessment methods.

We further propose EfRLFN, an efficient SR model with novel architectural innovations and a composite loss training strategy, supporting both video and image inputs. Extensive evaluation demonstrates EfRLFN's superior performance, delivering quality improvements and practical efficiency gains over existing methods. By combining carefully designed datasets, comprehensive benchmarking insights, and optimized model architecture, our work provides a complete foundation for advancing real-time SR research. Since EfRLFN is designed for single-image SR, exploring lightweight temporal extensions represents an important direction for future work. The released resources, including raw subjective data, enable future work in both super-resolution and perceptual quality assessment.

## 7 ACKNOWLEDGEMENTS

This work was supported by the The Ministry of Economic Development of the Russian Federation in accordance with the subsidy agreement (agreement identifier 000000C313925P4H0002; grant No 139-15-2025-012).

The research was carried out using the MSU-270 supercomputer of Lomonosov Moscow State University.

## 8 REPRODUCIBILITY STATEMENT

To ensure the reproducibility of our work, we have made the following resources available. The complete source code and pre-trained weights for the proposed EfRLFN model are included in the supplementary materials. A detailed description of the video collection and preprocessing pipeline from the dataset is provided in Section 4. Furthermore, the full set of experimental results, including additional ablations, can be found in Section 19.

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

## 9 APPENDIX CONTENTS

This appendix provides additional details, experiments, and analyses to support the main paper. Below is a summary of each section:

- **Section 10**: Discussion of use-cases, limitations, and potential future improvements.
- **Section 11**: Summary of real-time super-resolution (SR) models evaluated, including architectures and training configurations.
- **Section 12**: Comprehensive description of the experimental setup.
- **Section 13**: Analysis of video pairs sourced from YouTube in comparison to bicubic downscaled LR.
- **Section 14**: Full methodology and results of the crowd-sourced perceptual evaluation.
- **Section 15**: Ablation study on visual quality metrics and their impact on evaluations.
- **Section 16**: Ablation study on activation function replacements in the SPAN model.
- **Section 17**: Prompts and sampled queries used for dataset collection.

- **Section 18**: Extended visual comparisons of SR model outputs.

- **Section 19**: Expanded benchmark results (4× upscaling, additional metrics/datasets).

- **Section 20**: Dataset visualization and download links for the proposed StreamSR dataset.

## 10 DISCUSSION AND FUTURE WORK

**Computational Trade-offs.** Our benchmark intentionally excludes non-real-time super-resolution methods due to their low FPS. As visualized in Figure 1, state-of-the-art methods like RealESR-GAN (Wang et al., 2021) and SwinIR (Liang et al., 2021) achieve marginally better perceptual quality but fall far below the 30 FPS threshold required for real-time applications on consumer GPUs (NVIDIA RTX 2080 GPU, 720p to 1440p upscaling). For instance, Omni-SR, the fastest non-real-time method in our comparison, operates at 16 FPS - significantly slower than SPAN, which at 60 FPS was the slowest real-time method we tested.

**Codec Coverage Constraints.** The study's focus on YouTube-sourced video content naturally limits our evaluation to VP9, H.264, and AV1 codecs. While these account for most of web video traffic, specialized codecs used in professional settings (e.g., VVC) may exhibit different compression artifacts that could impact super-resolution performance.

**Societal Impact and Potential Misuses.** Like all enhancement technologies, our model carries dual-use risks, such as unauthorized upscaling of low-resolution surveillance footage beyond legal limits or generating synthetic high-resolution content from deliberately degraded sources.

**Future Work.** The real-time super-resolution technology offers significant benefits, including high-resolution streaming with lower bandwidth, which is particularly valuable for developing regions, but it also poses risks, where degraded content is enhanced to appear original. To support reproducibility and future research, we provide the complete implementation code, pretrained weights, validation datasets, and raw subjective evaluation scores as supplementary materials - this comprehensive package will enable other researchers to validate our findings, develop improved quality assessment methods, and advance real-time super-resolution techniques. Furthermore, the proposed StreamSR dataset and preference scores can be used to enhance existing super-resolution quality assessment techniques.

## 11 REAL-TIME SR MODELS SUMMARY

This section summarizes the real-time super-resolution models evaluated in our study. Table 5 provides a comparative analysis of their key specifications, including model size (parameters), computational complexity (FLOPs), supported upscaling factors, architectural design, and training datasets.

Table 5: Model Specifications of Real-Time SR Methods. Architectural details and scaling capabilities are shown.

| Method | Params (M) | FLOPs (G) | Upscale Factors | Architecture Type | Training Data |
|---|---|---|---|---|---|
| ESPCN (Talab et al., 2019) | 0.04 | 2.43 | 2×,3×,4× | Sub-pixel CNN | DIV2K |
| XLSR (Ayazoglu, 2021) | 0.03 | 1.81 | 4× | CNN | DIV2K |
| RLFN (Kong et al., 2022) | 0.40 | 136 | 2×,4× | CNN+Attention | DIV2K |
| AsConvSR (Guo et al., 2023) | 2.35 | 9 | 2× | Assembled Convolution | DIV2K+Flickr2K |
| Bicubic++ (Bilecen & Ayazoglu, 2023) | 0.05 | 0.83 | 3× | CNN | DIV2K |
| NVIDIA VSR (Choi, 2023) | - | - | 2×,3×,4× | CNN+Attention | Proprietary |
| RT4KSR (Zamfir et al., 2023) | 0.05 | 172 | 2× | CNN+Attention | DIV2K+Flickr2K |
| SAFMN (Sun et al., 2023) | 0.23 | 52 | 4× | CNN with Feature modulation | DIV2K+Flickr2K |
| SMFANet (Zheng et al., 2024) | 0.20 | 11 | 4× | CNN with Feature modulation | DIV2K+Flickr2K |
| SPAN (Wan et al., 2024) | 0.43 | 28 | 2×,4× | Parameter-free attention | DIV2K+Flickr2K |

Table 6: Performance Comparison of Training Strategies on DIV2K.

| Model | LR Type | PSNR ↑ | SSIM ↑ | LPIPS ↓ | ERQA ↑ |
|---|---|---|---|---|---|
| SPAN | Synthetic | 32.645 | 0.834 | 0.227 | 0.489 |
| | Real LR | **33.511** | **0.847** | **0.206** | **0.501** |
| RLFN | Synthetic | 32.718 | 0.845 | 0.193 | 0.517 |
| | Real LR | **33.816** | **0.855** | **0.182** | **0.535** |
| EfRLFN | Synthetic | 33.983 | 0.859 | 0.179 | 0.529 |
| | Real LR | **34.553** | **0.865** | **0.173** | **0.536** |

## 12 EXPERIMENTAL SETUP

We trained each model for 500 epochs, following the training processes described in the original papers. All models were evaluated on 720p videos with $2\times$ upscaling to 1440p resolution. For FPS measurements, we ran each model on an NVIDIA GeForce RTX 2080 GPU and calculated runtime on the same test sequence of 100 frames, averaged across 3 sequential runs. During EfRLFN training we set loss coefficients $\lambda_{\text{Charb}} = 1$, $\lambda_{\text{VGG}} = 10^{-3}$, and $\lambda_{\text{Sobel}} = 10^{-1}$ for normalization purposes.

## 13 THE IMPACT OF USING LR-HR PAIRS

SPAN, RLFN, and EfRLFN models trained using HR-LR pairs from our dataset show better performance (Table 6) compared to the same models trained using a bicubic downsampling process. In the latter case, the HR images from our dataset were downsampled to generate LR images using bicubic interpolation. This observation highlights the effectiveness of collecting domain-specific HR-LR pairs directly from YouTube, demonstrating that these pairs are more representative and beneficial than synthetic bicubic downsampling.

## 14 CROWD-SOURCED STUDY DETAILS

The evaluation was organized into two distinct tracks: one focusing on $2\times$ SR and the other on $4\times$ SR. Given the limited number of assessors with screen resolutions exceeding Full HD, we opted to present each test video as a center Full HD crop and showed videos sequentially.

During the experiment, participants were presented with pairs of videos generated by two randomly selected SR models. They were tasked with selecting the video that exhibited fewer visual artifacts; an option to indicate that the videos were "indistinguishable" was also provided. Each pair of videos was assessed by a total of 30 participants, with each participant evaluating 10 pairs of videos.

To ensure the integrity of the results, we included two verification questions with defined correct answers among the 10 pairs. Responses from any participant who failed to answer one or more of these verification questions correctly were excluded from further analysis. A total of 3,822 unique individuals participated in our subjective evaluation. The final subjective scores were derived from the 37,184 responses, applying the Bradley-Terry model (Bradley & Terry, 1952).

Figure 8 summarizes our setup for the study and visualizes the interface of Subjectify.us service used for crowd-sourcing.

Before completing the comparison, each participant was shown the following instructions:

> You will see video pairs shown one after another. For each pair, select the video with better visual quality (sharper, with fewer distortions like blurring or pixelation). If they look identical, choose "Indistinguishable."

> Be careful—there are verification questions in this test. Incorrect answers on verification checks will lead to rejection.

## 15 CORRELATION OF METRICS WITH SUBJECTIVE SCORES

Figure 7 presents the correlation analysis between objective video quality metrics and human subjective evaluations. Our experimental results demonstrate that four metrics – ClipIQA, ERQA, LPIPS, and MUSIQ – show significant correlations with subjective quality assessments.

These findings align with prior research in three key aspects: (1) the superior performance of feature-based metrics (LPIPS, MUSIQ) over traditional approaches, (2) ERQA's specialization for restoration quality assessment, and (3) the poor correlation of PSNR with human perception, consistent with other works (Huynh-Thu & Ghanbari, 2008; Bogatyrev et al., 2024).

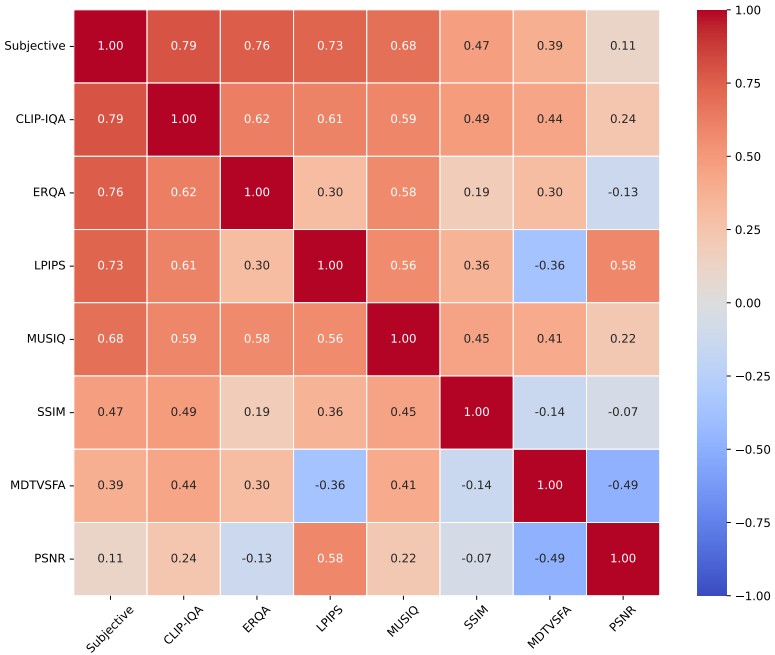

Figure 7: Pearson Correlation between objective quality metrics and subjective scores.

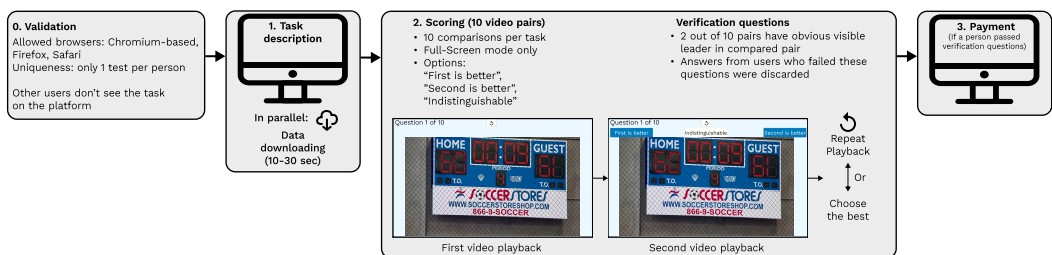

Figure 8: An overall scheme of the crowdsource subjective study described in Section 14. A screenshot of the interface of crowdsource subjective platform.

## 16 IMPROVING SPAN WITH HYPERBOLIC TANGENT ACTIVATION

Our experiments demonstrate that replacing the shifted sigmoid ($\text{Sigmoid}(x) - 0.5$) with hyperbolic tangent ($\tanh$) activation in SPAN leads to consistent improvements over the original formulation. Table 7 compares both variants across standard benchmarks:

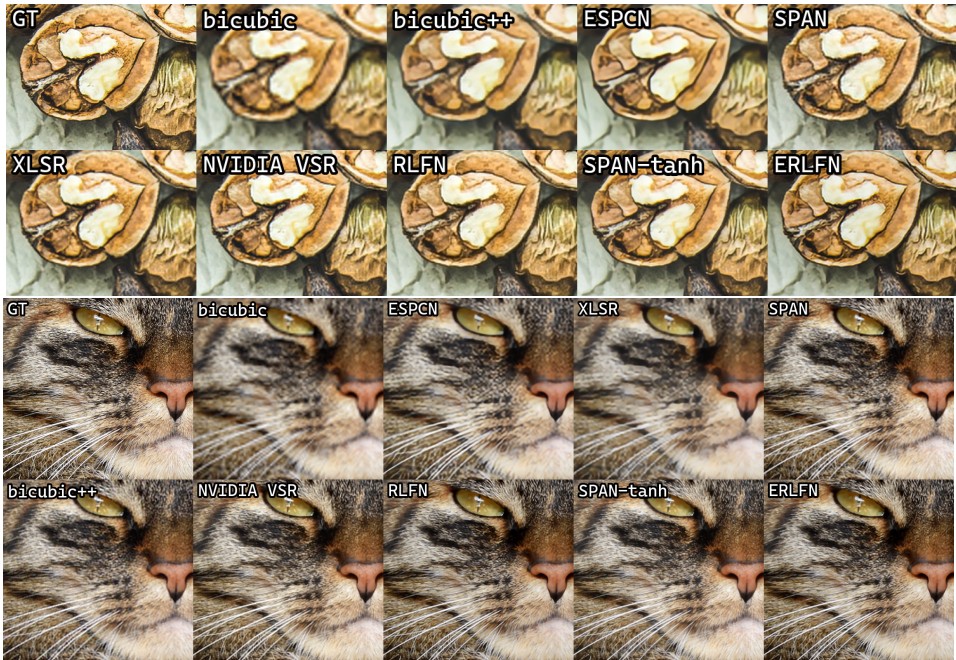

Figure 9: The comparison between several $2\times$ real-time SR models.

Table 7: Performance Comparison: SPAN ($\sigma = 0.5$) vs. SPAN (tanh).

| Model | Set14 SSIM↑/LPIPS↓ | BSD100 SSIM↑/LPIPS↓ | Urban100 SSIM↑/LPIPS↓ | DIV2K SSIM↑/LPIPS↓ |
|---|---|---|---|---|
| SPAN ($\sigma = 0.5$) | 0.836 / 0.142 | 0.837 / 0.239 | 0.847 / 0.118 | 0.890 / 0.175 |
| SPAN (tanh) | **0.837 / 0.140** | **0.841 / 0.228** | **0.853 / 0.115** | **0.891 / 0.167** |

## 17 LLM-GENERATED QUERIES

Table 9 presents the description of video categories used in our study, including their descriptions and the corresponding prompt templates used for query generation. Table 8 complements this by providing representative examples of LLM-generated search queries organized by these categories, demonstrating the diversity of content collected for our benchmark dataset.

## 18 VISUAL COMPARISON

Figure 9 compares real-time SR models on DIV2K. Figure 11 demonstrates significant quality gains after StreamSR fine-tuning, particularly in edge preservation and temporal stability.

## 19 FULL RESULTS

Table 12 presents comprehensive full results of our benchmark. Tables 10 and 11 provide full validation results on other datasets, including Set14 (Zeyde et al., 2012) and REDS (Nah et al., 2019).

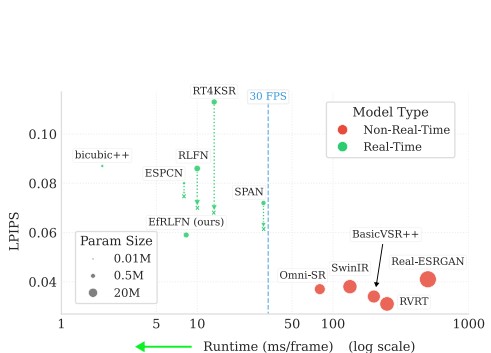

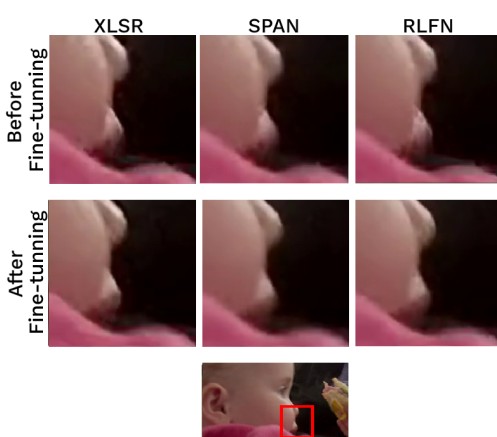

Figure 10: Trade-off between LPIPS and runtime speed for various 2× super-resolution models.

Figure 11: Visual quality comparison before and after fine-tuning on our dataset.

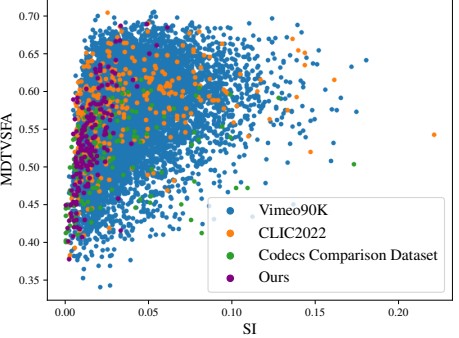

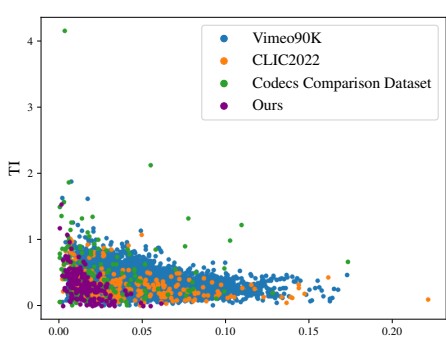

Figure 12: Distribution of Google Spatial and Temporal (Wang et al., 2019) information and MDTVSFA (Li et al., 2021a) for videos from different datasets.

Table 8: LLM-generated queries examples.

| Sports | Animals | Human Activities |
|---|---|---|
| 1. "highlights of football" | 1. "wildlife documentary" | 1. "people dancing" |
| 2. "sports slow motion" | 2. "cute animal videos" | 2. "cooking tutorials" |
| 3. "extreme sports action" | 3. "slow motion animals" | 3. "fitness workouts" |
| 4. "basketball dunks" | 4. "pets playing" | 4. "street performers" |
| 5. "soccer goals" | 5. "nature interactions" | 5. "DIY projects" |
| **Art and Crafts** | **Macro Videos** | **Historical Footage** |
| 1. "time-lapse painting" | 1. "macro insects" | 1. "historical footage" |
| 2. "sculpting techniques" | 2. "close-up flowers" | 2. "old documentaries" |
| 3. "DIY crafts" | 3. "macro techniques" | 3. "historical events" |
| 4. "artistic creations" | 4. "tiny world" | 4. "classic film clips" |
| 5. "watercolor tutorial" | 5. "everyday objects" | 5. "moon landing" |
| **Animated Content** | **Science** | **Nature/Landscapes** |
| 1. "animation films" | 1. "science experiments" | 1. "nature landscapes" |
| 2. "animated scenes" | 2. "physics demos" | 2. "time-lapse nature" |
| 3. "cartoon clips" | 3. "chemical reactions" | 3. "ocean waves" |
| 4. "stop motion" | 4. "biology explained" | 4. "forest scenery" |
| 5. "3D animation" | 5. "science phenomena" | 5. "mountain views" |

Table 9: YouTube Content Categories and Query Generation Prompts.

| Category Name | Category Definition | Query Generation Prompt |
|---|---|---|
| Nature and Landscapes | Scenic natural environments and geographical features. | "Generate 100 diverse YouTube search queries about nature and landscapes, including scenic views, wildlife habitats, waterfalls, forests, and time-lapse nature videos." |
| Urban Environments | Human-made structures and city landscapes. | "Provide 100 YouTube search queries related to urban environments, such as cityscapes, street photography, architectural marvels, urban exploration, and drone footage of cities." |
| Sports | Athletic activities and competitive physical games. | "List 100 YouTube search queries for sports content, including game highlights, athlete interviews, training routines, extreme sports, and sports analysis." |
| Animals | Living organisms from the animal kingdom. | "Suggest 100 YouTube search queries about animals, covering wildlife documentaries, pet care tips, rare animal sightings, funny animal compilations, and animal behavior studies." |
| Human Activities | Actions and behaviors performed by people. | "Generate 100 YouTube search queries focused on human activities, such as daily routines, cultural festivals, street performances, social experiments, and work-life documentaries." |
| Art and Crafts | Creative works and handmade objects. | "Create 100 YouTube search queries for art and crafts, including painting tutorials, DIY projects, sculpture making, digital art timelapses, and craft ideas for beginners." |
| Macro Videos | Extreme close-up footage of small subjects. | "Provide 100 YouTube search queries for macro videos, such as close-up insect footage, microscopic water droplets, detailed textures, and extreme close-up photography." |
| Historical Footage | Recorded material from past events. | "List 100 YouTube search queries for historical footage, including old war videos, vintage city life, archival documentaries, and restored historical clips." |
| Animated Content | Moving images created through animation techniques. | "Suggest 100 YouTube search queries for animated content, like short animated films, 3D animation breakdowns, cartoon compilations, and motion graphics tutorials." |
| Science | Illustrations of scientific principles. | "Generate 100 YouTube search queries about scientific demonstrations, including chemistry experiments, physics simulations, biology dissections, and engineering prototypes." |
| Technology Reviews | Evaluations of technological products. | "Provide 100 YouTube search queries for technology reviews, such as smartphone comparisons, gadget unboxings, software tutorials, and futuristic tech showcases." |
| Cooking and Recipes | Food preparation instructions and techniques. | "List 100 YouTube search queries for cooking and recipes, including easy meal prep, gourmet dishes, baking tutorials, street food tours, and kitchen hacks." |
| Travel Vlogs | Video blogs documenting travel experiences. | "Suggest 100 YouTube search queries for travel vlogs, covering backpacking adventures, luxury travel guides, hidden tourist spots, and cultural immersion experiences." |
| Fitness and Workouts | Physical exercise routines and training. | "Generate 100 YouTube search queries for fitness and workouts, like home exercise routines, gym training tips, yoga sessions, and body transformation stories." |
| Music Performances | Live or recorded musical presentations. | "Provide 100 YouTube search queries for music performances, including live concerts, acoustic covers, music festival highlights, and street musician videos." |
| Educational Tutorials | Instructional content for learning. | "List 100 YouTube search queries for educational tutorials, such as math problem-solving, language learning, coding lessons, and science explainers." |
| Gaming Content | Video game-related material. | "Suggest 100 YouTube search queries for gaming content, like walkthroughs, esports tournaments, game reviews, speedruns, and funny gaming moments." |
| DIY Projects | Do-it-yourself creative endeavors. | "Generate 100 YouTube search queries for DIY projects, including home improvement hacks, handmade crafts, upcycling ideas, and woodworking tutorials." |
| Drone Footage | Aerial video captured by drones. | "Provide 100 YouTube search queries for drone footage, such as aerial nature shots, city flyovers, mountain landscapes, and cinematic drone cinematography." |
| Time-lapse Videos | Accelerated video sequences showing change over time. | "List 100 YouTube search queries for time-lapse videos, including sunset transitions, city day-to-night changes, plant growth, and construction progressions." |

Table 10: Comparison of real-time SR models on popular benchmarks (2× upscaling). Best results in **bold**, second best underlined. "$T$" indicates fine-tuning on our dataset.

| Model | BSD100 PSNR↑/SSIM↑/LPIPS↓ | Urban100 PSNR↑/SSIM↑/LPIPS↓ | Set14 PSNR↑/SSIM↑/LPIPS↓ | REDS PSNR↑/SSIM↑/LPIPS↓ | DIV2K PSNR↑/SSIM↑/LPIPS↓ |
|---|---|---|---|---|---|
| RT4KSR | 28.1/0.805/0.296 | 27.3/0.769/0.244 | 28.9/0.782/0.231 | 26.7/0.758/0.268 | 30.2/0.860/0.221 |
| RT4KSR$^T$ | 28.3/0.812/0.254 | 27.5/0.766/0.223 | 29.1/0.789/0.210 | 26.9/0.765/0.245 | 30.5/0.864/0.186 |
| AsConvSR$^T$ | 28.7/0.832/0.270 | 28.1/0.824/0.208 | 29.5/0.801/0.205 | 27.3/0.781/0.238 | 30.8/0.883/0.206 |
| bicubic | 27.0/0.752/0.386 | 26.2/0.711/0.324 | 27.8/0.725/0.308 | 25.6/0.702/0.342 | 29.1/0.820/0.292 |
| bicubic++$^T$ | 27.3/0.768/0.360 | 26.5/0.725/0.298 | 28.1/0.741/0.285 | 25.9/0.718/0.318 | 29.4/0.832/0.270 |
| ESPCN | 24.5/0.492/0.534 | 23.8/0.505/0.437 | 25.2/0.518/0.432 | 23.4/0.487/0.478 | 26.1/0.514/0.515 |
| ESPCN$^T$ | 28.2/0.814/0.236 | 27.6/0.840/0.150 | 29.0/0.795/0.185 | 26.8/0.772/0.215 | 30.4/0.862/0.180 |
| XLSR | 27.9/0.788/0.243 | 27.2/0.786/0.152 | 28.7/0.772/0.178 | 26.4/0.751/0.208 | 30.1/0.858/0.180 |
| XLSR$^T$ | 28.4/0.817/0.244 | 27.8/0.828/0.158 | 29.2/0.801/0.183 | 26.9/0.779/0.216 | 30.5/0.864/0.183 |
| SMFANet$^T$ | 28.1/0.803/0.239 | 27.5/0.798/0.153 | 28.9/0.787/0.180 | 26.7/0.766/0.209 | 30.4/0.865/0.177 |
| SAFMN$^T$ | 28.4/0.813/0.236 | 27.8/0.808/0.150 | 29.2/0.797/0.178 | 27.0/0.777/0.207 | 30.7/0.871/0.175 |
| RLFN | 28.2/0.805/0.238 | 27.6/0.803/0.147 | 29.0/0.792/0.172 | 26.8/0.771/0.203 | 30.6/0.876/0.175 |
| RLFN$^T$ | 28.9/0.834/0.239 | 28.3/0.845/0.153 | 29.7/0.822/0.175 | 27.5/0.801/0.206 | 31.1/0.881/0.178 |
| SPAN | 28.9/0.836/0.222 | 28.4/0.841/0.139 | 29.8/0.819/0.155 | 27.6/0.798/0.186 | 31.2/0.887/0.168 |
| SPAN$^T$ | 29.0/0.837/0.239 | 28.5/0.847/0.118 | 29.9/0.826/0.170 | 27.7/0.805/0.205 | 31.3/0.890/0.175 |
| EfRLFN$^T$ | 29.3/0.847/0.190 | 28.8/0.856/0.116 | 30.2/0.835/0.148 | 28.0/0.814/0.178 | 31.5/0.892/0.145 |

Table 11: Comparison of real-time SR models on popular benchmarks (4× upscaling). Best results in **bold**, second best underlined. "$T$" indicates fine-tuning on our dataset.

| Model | BSD100 PSNR↑/SSIM↑/LPIPS↓ | Urban100 PSNR↑/SSIM↑/LPIPS↓ | Set14 PSNR↑/SSIM↑/LPIPS↓ | REDS PSNR↑/SSIM↑/LPIPS↓ | DIV2K PSNR↑/SSIM↑/LPIPS↓ |
|---|---|---|---|---|---|
| RT4KSR$^T$ | 24.8/0.409/0.575 | 23.7/0.489/0.521 | 25.1/0.452/0.503 | 23.5/0.471/0.542 | 27.3/0.696/0.489 |
| AsConvSR$^T$ | 25.1/0.433/0.586 | 24.0/0.500/0.512 | 25.4/0.468/0.488 | 23.8/0.489/0.518 | 27.6/0.701/0.471 |
| bicubic | 24.9/0.423/0.589 | 23.8/0.493/0.589 | 25.2/0.445/0.521 | 23.6/0.463/0.553 | 26.8/0.562/0.484 |
| ESPCN | 25.0/0.434/0.671 | 23.9/0.432/0.597 | 25.3/0.447/0.563 | 23.7/0.438/0.602 | 26.9/0.539/0.572 |
| ESPCN$^T$ | 25.2/0.442/0.662 | 24.1/0.452/0.585 | 25.5/0.459/0.548 | 23.9/0.445/0.587 | 27.1/0.551/0.557 |
| XLSR | 26.8/0.587/0.497 | 25.7/0.580/0.466 | 27.1/0.562/0.421 | 25.5/0.548/0.451 | 28.9/0.704/0.391 |
| XLSR$^T$ | 27.5/0.659/0.450 | 26.4/0.672/0.341 | 27.8/0.631/0.388 | 26.2/0.612/0.402 | 29.6/0.759/0.339 |
| SMFANet | 27.0/0.602/0.488 | 25.9/0.595/0.458 | 27.3/0.578/0.413 | 25.7/0.563/0.438 | 29.1/0.715/0.384 |
| SMFANet$^T$ | 27.2/0.612/0.482 | 26.1/0.608/0.450 | 27.5/0.588/0.407 | 25.9/0.573/0.432 | 29.3/0.722/0.380 |
| SAFMN | 27.3/0.622/0.475 | 26.2/0.620/0.442 | 27.6/0.601/0.402 | 26.0/0.586/0.425 | 29.4/0.729/0.376 |
| SAFMN$^T$ | 27.5/0.632/0.468 | 26.4/0.632/0.434 | 27.8/0.611/0.398 | 26.2/0.596/0.420 | 29.6/0.736/0.372 |
| RLFN | 27.1/0.603/0.492 | 26.0/0.597/0.460 | 27.4/0.578/0.409 | 25.8/0.563/0.434 | 29.2/0.721/0.386 |
| RLFN$^T$ | 27.9/0.674/0.445 | 26.8/0.688/0.336 | 28.2/0.647/0.382 | 26.6/0.632/0.398 | 30.0/0.774/0.334 |
| SPAN | 27.5/0.632/0.481 | 26.4/0.639/0.369 | 27.8/0.618/0.358 | 26.2/0.603/0.384 | 29.6/0.739/0.341 |
| SPAN$^T$ | 27.7/0.649/0.483 | 26.6/0.650/0.406 | 28.0/0.625/0.375 | 26.4/0.610/0.401 | 29.8/0.750/0.375 |
| **EfRLFN$^T$** | **28.2/0.682/0.429** | **27.1/0.699/0.327** | **28.5/0.658/0.365** | **26.9/0.643/0.381** | **30.3/0.778/0.331** |

Table 12: Ranking of real-time SR methods with 95% confidence intervals. Best results in **bold**. "T" indicates fine-tuning. N/A means that model does not have pretrained weights for a specific upscaling factor.

| Method | 2× Track | | | | | | | | 4× Track | | | | | | | |
|---|---|---|---|---|---|---|---|---|---|---|---|---|---|---|---|---|
| | Subj. | PSNR | SSIM | LPIPS | MUSIQ | CLIP-IQA | ERQA | MDTVSFA | Subj. | PSNR | SSIM | LPIPS | MUSIQ | CLIP-IQA | ERQA | MDTVSFA |
| AsConvSR$^T$ | 1.93±0.11 | 35.25±0.15 | 0.912±0.003 | 0.214±0.008 | 52.3±1.2 | 0.48±0.02 | 0.53±0.01 | 0.512±0.008 | 1.32±0.14 | 32.65±0.15 | 0.865±0.003 | 0.382±0.008 | 41.2±1.1 | 0.38±0.02 | 0.42±0.01 | 0.495±0.007 |
| RT4KSR | 2.40±0.12 | 36.45±0.16 | 0.918±0.003 | 0.213±0.009 | 54.1±1.1 | 0.49±0.02 | 0.54±0.01 | 0.518±0.008 | 1.40±0.14 | 32.65±0.16 | 0.866±0.003 | 0.382±0.009 | 39.8±1.0 | 0.36±0.02 | 0.41±0.01 | 0.492±0.007 |
| RT4KSR$^T$ | 2.43±0.11 | 37.55±0.14 | 0.925±0.002 | 0.070±0.003 | 58.7±1.3 | 0.53±0.02 | 0.58±0.01 | 0.527±0.008 | 1.45±0.14 | 32.65±0.14 | 0.867±0.002 | 0.382±0.003 | 42.1±1.3 | 0.39±0.02 | 0.43±0.01 | 0.498±0.007 |
| bicubic | 2.44±0.14 | 30.32±0.21 | 0.872±0.004 | 0.076±0.005 | 49.5±1.2 | 0.44±0.024 | 0.48±0.01 | 0.502±0.008 | 1.49±0.13 | 32.81±0.21 | 0.842±0.004 | 0.240±0.005 | 42.5±1.3 | 0.40±0.024 | 0.45±0.01 | 0.503±0.007 |
| ESPCN | 2.48±0.12 | 30.71±0.18 | 0.878±0.003 | 0.078±0.004 | 50.2±1.0 | 0.45±0.02 | 0.49±0.01 | 0.505±0.008 | 1.52±0.11 | 32.04±0.18 | 0.838±0.003 | 0.276±0.004 | 43.1±1.1 | 0.41±0.02 | 0.46±0.01 | 0.506±0.007 |
| ESPCN$^T$ | 2.49±0.11 | 35.76±0.15 | 0.908±0.002 | 0.072±0.003 | 56.9±1.2 | 0.51±0.02 | 0.55±0.01 | 0.521±0.008 | 1.96±0.15 | 32.08±0.15 | 0.848±0.002 | 0.215±0.003 | 45.3±1.2 | 0.44±0.02 | 0.48±0.01 | 0.512±0.007 |
| Bicubic++$^T$ | 2.44±0.13 | 36.79±0.15 | 0.915±0.002 | 0.087±0.004 | 57.2±1.3 | 0.52±0.02 | 0.56±0.01 | 0.523±0.008 | 1.86±0.13 | 33.12±0.15 | 0.852±0.002 | 0.265±0.004 | 40.5±1.1 | 0.39±0.02 | 0.47±0.01 | 0.509±0.007 |
| NVIDIA VSR | 2.57±0.14 | 37.40±0.13 | 0.922±0.002 | 0.082±0.003 | 60.1±1.4 | 0.56±0.02 | 0.60±0.01 | 0.534±0.008 | 2.31±0.12 | 33.55±0.13 | 0.858±0.002 | 0.207±0.003 | 47.8±1.3 | 0.47±0.02 | 0.50±0.01 | 0.518±0.007 |
| XLSR | N/A | N/A | N/A | N/A | N/A | N/A | N/A | N/A | 2.13±0.14 | 33.44±0.14 | 0.855±0.003 | 0.263±0.004 | 43.9±1.4 | 0.42±0.024 | 0.48±0.01 | 0.510±0.007 |
| XLSR$^T$ | 2.56±0.12 | 37.25±0.14 | 0.920±0.002 | 0.230±0.010 | 53.4±1.1 | 0.47±0.02 | 0.51±0.01 | 0.515±0.008 | 2.17±0.12 | 33.44±0.12 | 0.856±0.002 | 0.263±0.010 | 44.6±1.2 | 0.43±0.02 | 0.49±0.01 | 0.513±0.007 |
| SMFANet | N/A | N/A | N/A | N/A | N/A | N/A | N/A | N/A | 1.98±0.16 | 33.05±0.16 | 0.850±0.003 | 0.191±0.005 | 43.6±1.4 | 0.42±0.024 | 0.47±0.01 | 0.508±0.007 |
| SMFANet$^T$ | 2.37±0.14 | 37.14±0.15 | 0.919±0.002 | 0.158±0.007 | 56.4±1.2 | 0.51±0.02 | 0.55±0.01 | 0.520±0.008 | 2.02±0.14 | 33.06±0.14 | 0.851±0.002 | 0.183±0.007 | 44.3±1.2 | 0.43±0.02 | 0.48±0.01 | 0.511±0.007 |
| SAFMN | N/A | N/A | N/A | N/A | N/A | N/A | N/A | N/A | 1.91±0.17 | 33.36±0.17 | 0.853±0.003 | 0.204±0.005 | 43.4±1.3 | 0.41±0.024 | 0.46±0.01 | 0.505±0.007 |
| SAFMN$^T$ | 2.26±0.15 | 37.09±0.15 | 0.918±0.002 | 0.122±0.006 | 57.9±1.3 | 0.53±0.02 | 0.57±0.01 | 0.525±0.008 | 1.95±0.15 | 33.38±0.15 | 0.854±0.002 | 0.199±0.006 | 44.1±1.1 | 0.42±0.02 | 0.47±0.01 | 0.509±0.007 |
| RLFN | 2.17±0.15 | 37.03±0.15 | 0.917±0.002 | 0.086±0.004 | 59.3±1.3 | 0.54±0.02 | 0.58±0.01 | 0.529±0.008 | 2.24±0.16 | 34.42±0.15 | 0.862±0.002 | 0.247±0.004 | 43.9±1.1 | 0.42±0.02 | 0.49±0.01 | 0.515±0.007 |
| RLFN$^T$ | 2.69±0.13 | 37.63±0.12 | 0.924±0.002 | 0.072±0.003 | 62.4±1.4 | 0.58±0.02 | 0.62±0.01 | 0.537±0.008 | 4.32±0.13 | 33.82±0.12 | 0.860±0.002 | 0.182±0.003 | 49.3±1.4 | 0.51±0.02 | 0.52±0.01 | 0.522±0.007 |
| SPAN | 2.55±0.12 | 37.45±0.13 | 0.921±0.002 | 0.066±0.003 | 61.8±1.3 | 0.57±0.02 | 0.61±0.01 | 0.535±0.008 | 1.87±0.15 | 32.47±0.12 | 0.845±0.002 | 0.377±0.003 | 46.2±1.2 | 0.46±0.02 | 0.51±0.01 | 0.520±0.007 |
| SPAN$^T$ | 3.13±0.15 | 37.73±0.12 | 0.925±0.002 | 0.063±0.003 | 64.5±1.5 | 0.61±0.02 | 0.65±0.01 | 0.547±0.008 | 3.14±0.12 | 33.51±0.12 | 0.859±0.002 | 0.206±0.003 | 48.1±1.3 | 0.49±0.02 | 0.52±0.01 | 0.523±0.007 |
| **ERLFN$^T$** | **3.33±0.14** | **37.85±0.11** | **0.928±0.002** | **0.059±0.003** | **67.8±1.4** | **0.65±0.02** | **0.74±0.01** | **0.547±0.008** | **4.52±0.13** | **34.55±0.11** | **0.865±0.002** | **0.173±0.003** | **52.7±1.5** | **0.58±0.02** | **0.539±0.01** | **0.527±0.007** |
| *Non-real-time SR models* | | | | | | | | | | | | | | | | |
| Real-ESRGAN | 3.87±0.13 | 37.65±0.12 | 0.926±0.002 | 0.240±0.007 | 68.1±1.2 | 0.66±0.02 | 0.74±0.02 | 0.5485±0.008 | - | - | - | - | - | - | - | - |
| SwinIR | 4.33±0.15 | 37.88±0.11 | 0.928±0.003 | 0.224±0.010 | 69.5±1.4 | 0.67±0.03 | 0.75±0.01 | 0.5505±0.008 | - | - | - | - | - | - | - | - |
| Omni-SR | 4.50±0.14 | 37.92±0.15 | 0.929±0.002 | 0.218±0.006 | 70.2±1.2 | 0.68±0.01 | 0.76±0.01 | 0.5525±0.009 | - | - | - | - | - | - | - | - |
| BasicVSR++ | 4.87±0.13 | 38.05±0.13 | 0.930±0.002 | 0.203±0.007 | 71.8±1.1 | 0.70±0.02 | 0.77±0.01 | 0.5885±0.007 | - | - | - | - | - | - | - | - |
| RVRT | 4.98±0.14 | 38.12±0.13 | 0.931±0.003 | 0.183±0.008 | 72.4±1.2 | 0.71±0.02 | 0.78±0.02 | 0.5625±0.008 | - | - | - | - | - | - | - | - |

## 20  DATASET PREVIEW

Figure 13 presents representative frames from the proposed StreamSR dataset. These samples illustrate the diversity of motion patterns, scene types, and compression characteristics captured in the data.

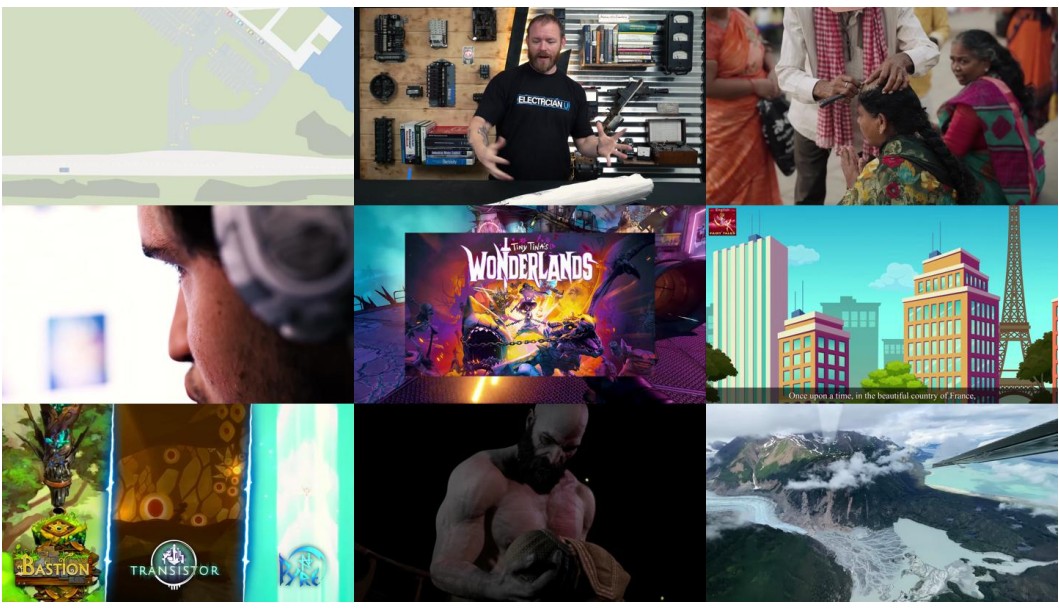

Figure 13: Example frames from the proposed StreamSR dataset.

