# OpenReview forum: "Exploring Real-Time Super-Resolution: Benchmarking and Fine-Tuning for Streaming Content"
_ICLR.cc/2026/Conference — ICLR 2026 Poster_

### Official Review · Reviewer_LP9y · 2025-10-29

**Soundness:** 3
**Presentation:** 2
**Contribution:** 2
**Rating:** 4
**Confidence:** 3

**Summary:**

The paper introduces StreamSR, a large-scale dataset of 5,200 compressed YouTube videos for realistic benchmarking of real-time super-resolution. It also proposes EfRLFN, an efficient model using tanh activations, channel attention, and a composite loss for better quality-speed trade-offs. Extensive experiments and a large user study show EfRLFN achieves the best performance among real-time SR models and that fine-tuning on StreamSR boosts other models’ results.

**Strengths:**

1. StreamSR focuses on real-world compressed videos instead of the clean, synthetic data used in most existing benchmarks.

2. The paper includes both objective metrics and a large-scale subjective user study, which adds credibility to the results.

3. In addition to the compressed dataset, this paper proposes EfRLFN, an efficient real-time super-resolution model designed to further enhance both visual quality and inference speed.

**Weaknesses:**

1. The proposed EfRLFN model is a very incremental change over the existing RLFN . It swaps an attention block (ESA for ECA) and an activation function (ReLU for tanh).

2. The paper frames the problem as super-resolution for streaming content (i.e., video), but the proposed model is a single-image SR model applied frame-by-frame. This approach completely ignores temporal information, which is a critical component of video processing.

Detailed comments:
 1. My main concern is the disconnect between the problem -- video streaming-- and the proposed solution, single-image super-resolution (SR). Processing each video frame independently can lead to temporal artifacts such as flickering or popping because the model lacks context from previous or future frames. A true video SR model would utilize this temporal information to maintain consistency. The paper justifies its approach by citing that existing VSR models are too slow, which is valid, but it does not address potential quality trade-offs of their method. When applying EfRLFN to video, does it introduce temporal instability compared to models like NVIDIA's VSR—which, although proprietary, likely incorporates temporal awareness?

 2.Furthermore, the paper's own results show that a large part of the performance gain comes from the data, not the model. When SPAN and RLFN are fine-tuned on StreamSR, their performance jumps dramatically. This is great validation for the dataset, but it makes the EfRLFN model itself seem less essential. It seems the paper's primary finding is "training on real-world compressed video is better," which, while true, is not a huge surprise

**Questions:**

Refer to above.

---

> ### Author Response · Authors · 2025-11-24
>
> Thank you for your review and thoughtful feedback. We would like to address each point below.
>
> >  My main concern is the disconnect between the problem -- video streaming-- and the proposed solution, single-image super-resolution (SR). Processing each video frame independently can lead to temporal artifacts such as flickering or popping because the model lacks context from previous or future frames.
>
> > When applying EfRLFN to video, does it introduce temporal instability compared to models like NVIDIA's VSR—which, although proprietary, likely incorporates temporal awareness?
>
> We agree that temporal consistency is important for video applications. Our choice of single-image SR is motivated by the strict latency requirements of real-time streaming: state-of-the-art multi-frame methods (e.g., BasicVSR++, RVRT) operate far below real-time speed for 360p inputs and therefore cannot be deployed in this setting.
>
> To directly address your question about temporal instability: we explicitly evaluated this through a large-scale user study where participants compared videos. As shown in Figure 5(a), EfRLFN was preferred over NVIDIA's VSR in 77.4% of cases. Since this evaluation was performed on videos, the results strongly indicate that any potential temporal artifacts in EfRLFN's output are, in practice, less perceptible or objectionable to viewers than those in NVIDIA's VSR’s output.
>
> > The proposed EfRLFN model is a very incremental change over the existing RLFN . It swaps an attention block (ESA for ECA) and an activation function (ReLU for tanh).
>
> > Furthermore, the paper's own results show that a large part of the performance gain comes from the data, not the model.
>
> StreamSR plays a central role in improving performance across all real-time SR models evaluated. At the same time, EfRLFN incorporates targeted architectural and loss-level choices (Sections 3.1-3.2) that consistently improve performance beyond what dataset adaptation alone provides:
> * Tanh activation preserves contrast in compressed regions better than ReLU and improves reconstruction of blocky/blurred textures (Fig. 5(b));
> * ECA yields more effective channel selection at negligible cost;
> * Composite loss (Charbonnier + VGG + Sobel) stabilizes optimization and reduces ringing and block-boundary artifacts (Fig. 6(a)).
>
> Table 3 shows that removing ECA or tanh consistently reduces performance, and Fig. 1 demonstrates the resulting improvement in the quality-runtime frontier. Thus, while StreamSR produces strong gains for all models, EfRLFN establishes a new Pareto-efficient point in the accuracy-efficiency trade-off.
>
> We agree that EfRLFN is not intended to introduce fundamentally new architecture. Rather, its goal is to combine lightweight yet effective design choices — many of which are under-utilized in real-time SR — to improve reconstruction quality for compressed video while maintaining real-time throughput. We adjusted the Introduction in the revised paper to reflect this pragmatic design philosophy more clearly.

---

> > ### Comment · Reviewer_LP9y · 2025-11-28
> > **Comment**
> >
> > Thank you for a detailed rebuttal. The rebuttal sufficiently addresses my concerns regarding the temporal instability. Therefore, I'm leaning towards accepting and will update my final rating.

---

> > > ### Author Response · Authors · 2025-11-28
> > >
> > > Thank you for your thorough review. We are pleased that our revisions and clarifications successfully addressed your feedback and led to a positive change.

---

### Official Review · Reviewer_2ofZ · 2025-10-31

**Soundness:** 3
**Presentation:** 3
**Contribution:** 2
**Rating:** 4
**Confidence:** 5

**Summary:**

This paper identifies a critical gap in existing super-resolution research: the lack of datasets that realistically model the compression artifacts found in real-world video streaming. To address this, the authors make several contributions. First, they introduce StreamSR, a new, large-scale dataset of 5,200 videos sourced from YouTube, which contains authentic compression artifacts. Second, they conduct a comprehensive benchmark of 11 state-of-the-art (SOTA) real-time SR models on this new dataset, notably including a large-scale subjective user study with over 3,800 participants. Third, they propose EfRLFN, a lightweight SR model based on RLFN, which incorporates modifications like Efficient Channel Attention (ECA) and a tanh activation function to achieve a strong balance of quality and runtime performance. The authors show that their model, as well as other models fine-tuned on StreamSR, achieve significant performance gains.

**Strengths:**

Dataset Significance and Quality: The paper's most valuable contribution is the StreamSR dataset. The authors correctly argue that popular datasets like DIV2K and Vimeo90K do not adequately represent the challenges of real-world streaming media. The effort to collect, filter, and curate a 5,200-video dataset with associated low- and high-resolution pairs from a streaming source is substantial and addresses a clear need in the community .

Comprehensive Benchmarking: The second major strength is the systematic and thorough benchmarking of 11 real-time SR methods. This evaluation is rigorous, employing 7 different objective metrics and, most importantly, a large-scale subjective study.
Large-Scale Subjective Study: The inclusion of a subjective evaluation with 3,822 participants and over 37,000 valid responses is highly commendable. This provides a much-needed perceptual grounding for the objective metrics, and the community is well-aware of the significant time and financial cost required to obtain such data. The paper's finding that fine-tuning on StreamSR significantly improves the performance of existing models like SPAN and RLFN strongly validates the dataset's utility and "real-world" nature.

**Weaknesses:**

Limited Architectural Novelty: The primary weakness is the limited originality of the proposed EfRLFN model. The architecture appears to be a thoughtful and effective combination of existing lightweight techniques rather than a novel design. The key modifications—such as replacing ESA with ECA and using a tanh activation—are well-motivated by prior work (e.g., SPAN ) and common practices from recent NTIRE challenges. While the ablation studies (e.g., Table 3, Figure 5(b)) are detailed and confirm the authors' design choices, they largely reinforce existing knowledge (e.g., the benefit of odd symmetric activation functions) rather than providing fundamentally new insights for the field.

Insufficient Evidence for "Hardware-Friendly" Claims: The paper repeatedly emphasizes that EfRLFN's design is "efficient"  and "hardware-friendly". However, these claims are supported only by FPS measurements from PyTorch on an NVIDIA RTX 2080 GPU. This benchmark is a useful proxy but does not represent the performance in a true deployment scenario. Real-world applications typically involve inference engines (e.g., TensorRT, OpenVINO) and further optimizations like model quantization, operator fusion, and platform-specific kernel tuning. I understand that a full deployment analysis is a complex task, but the "hardware-friendly" claims are not sufficiently substantiated by the current experiments. At a minimum, this limitation should be discussed in the paper.

**Questions:**

Reframing of Contributions: Given that the paper's most significant and lasting contribution is arguably the comprehensive StreamSR dataset and the large-scale benchmark, would the authors consider reframing the Abstract and Introduction? Currently, the EfRLFN model is presented as a primary contribution on par with the dataset. However, as noted, the model's architectural novelty is limited, and similar designs are common in NTIRE reports. Perhaps positioning the dataset and benchmark as the central contribution, with EfRLFN serving as a strong new baseline developed for this benchmark, would more accurately reflect the paper's impact.

Clarification of Scope (SISR vs. VSR): The paper's motivation is "video streaming" and "video content". However, the proposed EfRLFN model is explicitly a single-image SR (SISR) method ("we focus on image SR for our approach"). This is a valid and common strategy for real-time applications, but it does not leverage temporal information. Could the authors please clarify this distinction more prominently in the Abstract and Introduction? This would help set reader expectations, as "video super-resolution" often implies temporal (VSR) methods.
Deployment Performance: Following up on the weakness regarding efficiency claims: I would be very willing to reconsider my score if the authors could provide even preliminary data on deployment. For instance, what is the inference speed, throughput, or power consumption of EfRLFN and a key competitor (like SPAN or RLFN) when exported (e.g., to ONNX) and run on an inference engine like TensorRT or OpenVINO? This would provide much stronger evidence for the model's practical, "hardware-friendly" efficiency beyond the PyTorch runtime. If this is not feasible, I strongly suggest adding a discussion of this limitation and positioning platform-specific optimization as future work.

---

> ### Author Response · Authors · 2025-11-24
>
> Thank you for the detailed assessment and constructive suggestions. Below we address each point raised.
>
> > Deployment Performance: Following up on the weakness regarding efficiency claims: I would be very willing to reconsider my score if the authors could provide even preliminary data on deployment. For instance, what is the inference speed, throughput, or power consumption of EfRLFN and a key competitor (like SPAN or RLFN) when exported (e.g., to ONNX) and run on an inference engine like TensorRT or OpenVINO?
>
> To complement PyTorch benchmarks, we exported EfRLFN, RLFN, and SPAN to ONNX and evaluated them using ONNX Runtime with both CUDA and TensorRT execution providers. All models were tested on 480×360 inputs for both 2× and 4× upscaling.
>
> **2× upscaling:**
> | Model         | CUDA (ms) | TensorRT (ms) | Speedup | FPS (TRT) | Subjective Score |
> | ------------- | --------- | ------------- | ------- | --------- | --- |
> | RLFN          | 30.76     | 29.12         | 1.06×   | 34.3      | 2.69 ± 0.15 |
> | SPAN          | 16.29     | 10.46         | 1.55×   | 95.6      | 2.55 ± 0.12 |
> | EfRLFN (ours) | 17.43     | 12.07         | 1.44×   | 82.8      | 3.33 ± 0.14 |
>
> **4× upscaling:**
> | Model         | CUDA (ms) | TensorRT (ms) | Speedup | FPS (TRT) | Subjective Score |
> | ------------- | --------- | ------------- | ------- | --------- | --- |
> | RLFN          | 38.24     | 32.48         | 1.18×   | 30.8      | 4.32 ± 0.13 |
> | SPAN          | 17.20     | 10.86         | 1.58×   | 92.1      | 3.14 ± 0.12 |
> | EfRLFN (ours) | 20.25     | 14.66         | 1.38×   | 68.2    | 4.52 ± 0.13 |
>
>
> EfRLFN achieves lower latency than RLFN on both backends and exceeds real-time throughput (≥30 FPS). The architecture also benefits substantially from TensorRT optimizations, confirming its alignment with deployment-oriented execution engines. We added these findings to the revised paper in Section 5.2.
>
> > Clarification of Scope (SISR vs. VSR): The paper's motivation is "video streaming" and "video content". However, the proposed EfRLFN model is explicitly a single-image SR (SISR) method ("we focus on image SR for our approach").
>
> EfRLFN is intentionally a single-image SR model. As noted, real-time streaming scenarios impose strict latency constraints, and contemporary video SR methods that use temporal cues (e.g., BasicVSR++, RVRT) operate far below real-time speeds (Table 2). In the revised paper we updated the Introduction to explicitly state that our work focuses on SISR for real-time deployment, and that exploring lightweight temporal methods is an important direction for future work.
>
> > The primary weakness is the limited originality of the proposed EfRLFN model. The architecture appears to be a thoughtful and effective combination of existing lightweight techniques rather than a novel design.
>
> > Given that the paper's most significant and lasting contribution is arguably the comprehensive StreamSR dataset and the large-scale benchmark, would the authors consider reframing the Abstract and Introduction?
>
> EfRLFN is designed with the specific goal of achieving strong performance under strict real-time constraints on compressed streaming content. While the individual components (ECA, tanh, composite loss) only the incremental improvements, Sections 3.1-3.2 and Table 3 show that each contributes measurably to performance:
>
> * Odd-symmetric tanh activation improves local contrast restoration relative to ReLU or shifted sigmoid, particularly in regions affected by blocking and over-smoothing (Fig. 5(b)).
> * ECA selectively enhances channel interactions with negligible computational overhead.
> * Composite loss (Charbonnier + VGG + Sobel) stabilizes training and suppresses compression-induced ringing/edges, outperforming RLFN’s contrastive loss and all individual components (Fig. 6(a)).
>
> Removing any of these elements leads to a reduction in SSIM/LPIPS (Table 3). The combination yields a new Pareto-efficient point in the quality–efficiency trade-off (Fig. 1), which is central for real-time streaming SR. However, we do agree that the StreamSR dataset and benchmark constitute the most substantial long-term contribution. We revised the text to emphasize that EfRLFN is proposed as a strong, practically motivated baseline accompanying the dataset, rather than a fundamentally new architecture.
>
>
> If you have any additional suggestions for evaluating the “hardware-friendly” aspects of our model, we would be happy to address them. We would greatly appreciate it if you would consider raising your score.

---

> ### Author Response · Authors · 2025-11-28
>
> Thank you for your effort in reviewing our work and for your suggestions, which have helped make our paper better. Despite the reversion of the discussion, we are glad that our revision addressed your questions and led to a positive reassessment.

---

### Official Review · Reviewer_yoXt · 2025-10-31

**Soundness:** 3
**Presentation:** 2
**Contribution:** 2
**Rating:** 4
**Confidence:** 4

**Summary:**

This paper addresses real-time super-resolution for compressed streaming videos. The authors propose **StreamSR**, a large-scale dataset of real YouTube videos with compression artifacts, and **EfRLFN**, an efficient network designed for high-quality, low-latency upsampling. A comprehensive benchmark on 11 methods, including objective and subjective evaluations, demonstrates the effectiveness of the propsed approach.

**Strengths:**

1. The paper addresses a timely and practical problem: enhancing low-quality compressed videos in real-world streaming scenarios. By introducing both a new dataset and benchmark, it highlights the limitations of current SR methods under realistic conditions.
2. The StreamSR dataset fills an important gap by providing a large-scale collection of real YouTube videos with natural compression artifacts, offering a more realistic evaluation platform than synthetic benchmarks.
3. EfRLFN achieves competitive performance among real-time SR models, with strong results on the proposed dataset and favorable efficiency-quality trade-offs, validated through extensive experiments.

**Weaknesses:**

1. The paper focuses on real-time SR for compressed streaming videos, but the related work primarily discusses general real-time SR methods, with little discussion of existing compressed VSR approaches. Important works in this domain—such as *Learning Degradation-Robust Spatiotemporal Frequency-Transformer for Video Super-Resolution*—are not adequately reviewed. A dedicated discussion on compressed VSR literature is needed.
2. The main claimed contribution is the StreamSR dataset, yet no sample frames or metadata (e.g., compression settings like bitrates, codecs, and resolution scaling methods) are provided in the paper or supplementary material. Including visual examples and detailed encoding information would help reviewers assess the dataset’s quality and realism.
3. The proposed EfRLFN lacks clear architectural novelty for handling compression artifacts. The design appears to be a refined version of RLFN without explicit mechanisms to address compression distortions (e.g., blocking, blurring). The performance gain may stem largely from training on the new dataset rather than model innovation

**Questions:**

see weaknesses

---

> ### Author Response · Authors · 2025-11-24
>
> Thank you for the constructive feedback. We would like to address each point below.
>
> > Important works in this domain—such as Learning Degradation-Robust Spatiotemporal Frequency-Transformer for Video Super-Resolution—are not adequately reviewed.
>
> > A dedicated discussion on compressed VSR literature is needed.
>
> We appreciate the suggestion to expand the discussion of compression-aware video SR. In the 9-page limit, we focused on real-time and efficient SR methods, which are essential for our research. However, we agree that compressed-domain VSR is highly relevant, so we added a dedicated subsection in the revised version of the paper.
>
> We also evaluated FTVSR (proposed in *Learning Degradation-Robust Spatiotemporal Frequency-Transformer for Video Super-Resolution*) on StreamSR for completeness. Its inference speed is ~0.7 FPS for 2× upscaling of 720p inputs, which places it far outside our real-time threshold. Nevertheless, we discussed its design and relevance more clearly in the revised related-work section.
>
> > The main claimed contribution is the StreamSR dataset, yet no sample frames or metadata (e.g., compression settings like bitrates, codecs, and resolution scaling methods) are provided in the paper or supplementary material.
>
> A sample of the dataset (including training/test splits and video metadata) is available anonymously on Google Drive. The access link is provided in the README.md file within our supplementary materials. We agree, however, that presenting representative frame samples in the main paper would improve clarity. In the revised version of the paper, we included several visual examples in Section 12 of the Appendix.
>
> Regarding resolution scaling methods: we did not downscale high-resolution videos to create low-resolution counterparts. Instead, we downloaded aligned pairs of videos at different resolutions directly from YouTube, preserving YouTube’s original downscaling pipeline.
>
> > The design appears to be a refined version of RLFN without explicit mechanisms to address compression distortions (e.g., blocking, blurring). The performance gain may stem largely from training on the new dataset rather than model innovation
> EfRLFN is designed for strict real-time deployment, where incorporating heavy artifact-removal modules is infeasible.
>
> While training on our StreamSR dataset largely affected the performance of our model, the architectural modifications and loss selection also played a critical role. We performed an extensive ablation study on every component of our model and the proposed dataset. We identified lightweight architectural and loss-function components that provide tangible robustness to compression distortions:
> * Tanh activation preserves local contrast better than shifted sigmoid or ReLU, improving recovery of blocked/blurry regions (see Fig. 5(b)).
> * ECA attention improves channel-wise feature selection with negligible computational cost.
> * Composite loss (Charbonnier + VGG + Sobel) stabilizes training and suppresses ringing/blocking artifacts; Fig. 6(a) shows that removing any component degrades SSIM throughout training.
>
> Table 3 demonstrates that omitting ECA or tanh reduces performance, indicating each contributes beyond dataset effects alone. While EfRLFN does not introduce a completely new architecture, these targeted modifications result in a new Pareto-efficient point in the speed-quality trade-off, which is critical for real-time streaming SR.
>
>
> We believe the above clarifications and revisions address the raised concerns and strengthen both transparency and technical clarity. We hope this resolves the remaining questions and supports a positive reassessment of the submission.

---

> > ### Comment · Reviewer_yoXt · 2025-11-27
> >
> > Thanks for reviewer's comment, which addressed my concerns a lot. I raise my score to 6.

---

> > > ### Author Response · Authors · 2025-11-28
> > >
> > > We sincerely appreciate the time and care you took in evaluating our work. Although the discussion was reverted, we are glad that our clarifications and revisions resolved your concerns and resulted in a positive reassessment.

---

### Official Review · Reviewer_9P2C · 2025-10-31

**Soundness:** 3
**Presentation:** 3
**Contribution:** 3
**Rating:** 6
**Confidence:** 3

**Summary:**

This paper addresses the challenge of enhancing video streaming quality via super-resolution (SR) in real-time. The authors note that standard SR models trained on datasets like DIV2K or Vimeo-90K struggle with heavily compressed streaming videos (e.g. YouTube content), which introduce artifacts (blockiness, blur, loss of detail) not represented in those datasets. To bridge this gap, the paper introduces StreamSR, a new dataset of 5,200 YouTube video clips (over 10 million frames) spanning diverse genres and resolutions (360p–1440p) representative of real streaming scenarios. They benchmark 11 state-of-the-art real-time SR models on StreamSR under two upscaling tracks (2× and 4×), using 7 image quality metrics (including PSNR, SSIM, LPIPS, CLIP-IQA, etc.) and a large-scale user study with 3,800+ participants. In addition, the authors propose a new efficient SR model called EfRLFN (Efficient Residual Lightweight Feature Network), which builds upon the prior RLFN architecture with targeted improvements. EfRLFN integrates Efficient Channel Attention (ECA) and hyperbolic tangent activations (tanh), along with a refined training strategy, to achieve better image quality and faster inference. Empirically, EfRLFN achieves state-of-the-art performance among real-time methods on the StreamSR benchmark, delivering favorable quality–complexity trade-offs. Notably, fine-tuning existing SR models on the StreamSR data yields significant performance gains that also generalize to standard benchmarks (the fine-tuned models show improved metrics on Set14, Urban100, DIV2K, etc.). Overall, the paper provides a comprehensive evaluation of real-time SR in streaming contexts: the StreamSR dataset, an improved SR model (EfRLFN), and thorough experiments demonstrating both objective improvements (e.g. higher PSNR/SSIM) and subjective gains (user preference overwhelmingly favoring EfRLFN outputs).

**Strengths:**

1. Novelty: The new StreamSR dataset is a valuable contribution, addressing a need for training/evaluating SR on real-world streaming content. It comprises 5.2K YouTube videos (25–30s clips) with aligned low/high-resolution pairs at 360p→1440p (4×) and 720p→1440p (2×) scales. Unlike prior SR datasets which often use pristine images synthetically downscaled (e.g. DIV2K, REDS), StreamSR provides naturally compressed low-resolution frames containing authentic streaming artifacts (from common codecs like VP9, H.264, AV1). This yields a broader quality range (MDTVSFA 0.41–0.61) more reflective of real YouTube content. The dataset spans diverse genres and content types, and the authors carefully filtered and clustered videos to ensure diversity in the test set. Overall, StreamSR is novel and well-constructed, better representing real streaming scenarios than previous SR data (which were shorter or lacked compression artifacts). This greatly enhances the relevance of benchmark results and will likely support future research in streaming video SR.

2. Empirical Performance and Benchmarking Methodology: The experimental evaluation in this paper is comprehensive and rigorous. The authors benchmark 11 SR methods that are capable of real-time upscaling, ranging from classic baselines (Bicubic, ESPCN) to recent NTIRE winners (SMFANet, SPAN, RLFN) and even NVIDIA’s proprietary VSR solution. Each model is evaluated before and after fine-tuning on StreamSR, which provides insight into both out-of-the-box performance and achievable gains with adaptation. They report results on seven objective metrics, covering distortion-oriented measures (PSNR, SSIM), learned perceptual metric (LPIPS), and no-reference quality indices (e.g. CLIP-IQA and MUSIQ for human-aligned scoring), ensuring a well-rounded assessment of quality. Impressively, the paper also includes a user study with ~3,800 participants, collecting pairwise preference judgments. This large subjective test adds strong evidence of real-world efficacy, beyond just numbers. The results show EfRLFN’s outputs are overwhelmingly preferred by users over other methods (e.g. 77.4% of users favor EfRLFN over NVIDIA VSR in head-to-head comparisons). Such a combination of objective and subjective evaluation demonstrates empirical rigor. The benchmarking methodology is solid: the authors detail the evaluation protocol, use statistically significant comparisons (reporting confidence intervals), and even test generalization by applying StreamSR-trained models to standard datasets (Set14, Urban100, etc.). Notably, fine-tuning on StreamSR yields consistent improvements on these benchmarks, confirming the real-world data has broader benefits. Reproducibility is also addressed, the paper promises to release the dataset, code, and even the collected user study data, which should enable others to verify and build upon these results. Overall, the experiments are thorough and convincing, lending credibility to the paper’s claims.

**Weaknesses:**

1. Generalization Concerns: While StreamSR is a strong step toward real-world data, there are some questions about generalization. The dataset and experiments focus solely on YouTube videos and the prevalent codecs used there (VP9, H.264, AV1). This covers a large portion of online content, but it may not generalize to other domains or compression formats. For example, professional streaming or broadcast uses codecs like HEVC or VVC; these might produce different artifacts, and it’s unclear if models fine-tuned on StreamSR would handle them well. Similarly, the resolution range is up to 1440p, the benchmark does not evaluate 1080p→4K (2160p) upscaling, a common high-end scenario. The authors do show that models trained on StreamSR generalize to standard SR test sets, which is encouraging. However, the reverse direction (generalizing to content outside the YouTube distribution) remains uncertain. In future work, it would strengthen the contribution to expand codec and content coverage, e.g. include other video sources or higher-resolution tracks, to ensure the approach is robust beyond the exact domain it was trained on.

2. Missing Ablations and Comparisons: The authors conducted several ablation studies (activation functions, loss functions, etc.), but a few comparisons are either missing or left to supplemental material. For instance, an ablation on the composite loss would be useful to see how each component (Charbonnier, VGG perceptual, Sobel) contributes to final quality, the paper mentions this composite greatly helped training stability, so showing a breakdown of its impact would solidify that claim. Similarly, since EfRLFN is built upon RLFN, it would be enlightening to have a direct comparison of training with vs. without the new StreamSR data for that model: e.g. how much does fine-tuning RLFN on StreamSR improve its performance on compressed videos (the authors do include this result) and on standard benchmarks (they report gains). Such comparisons are partly given (Table 2 and discussion), but a more detailed ablation could isolate the effect of the dataset itself on a model’s performance. Another missing aspect is any comparison (even qualitative) to video-specific SR methods that leverage temporal redundancy. While most such methods are too slow for real-time, it would be interesting to see if, for example, a two-frame fusion approach could outperform single-frame EfRLFN at the cost of some speed, this could inspire future research on reaching a better quality/speed middle ground. Lastly, the main paper emphasizes 2× upscaling results; the 4× track results are deferred to the appendix. It might improve the paper to at least summarize the 4× findings in the core text, since 4× (360p→1440p) is a significant challenge and presumably where EfRLFN’s advantages also shine. In conclusion, additional ablations (loss function components, dataset contribution) and broader comparisons (to non-real-time or multi-frame methods) would further strengthen the work. These omissions do not undermine the existing results, but addressing them could provide a fuller picture and preempt questions from readers. The authors are encouraged to incorporate such analyses either in a revision or future follow-up, to solidify the paper’s conclusions and explore the boundaries of the proposed approach.

3. Methodological Clarity and Experimental Limitations: For the most part the paper is well-written and thorough, but a few methodological points could be clearer. The novelty of the EfRLFN model might be seen as somewhat incremental, it is essentially an improved variant of RLFN with known techniques (ECA attention, tanh activation) adopted from prior works. The paper does justify these choices well and supports them with ablations, but some readers may feel the model innovation is an optimization of existing components rather than a fundamentally new architecture. This is not a severe issue (pragmatic improvements are valuable for real-time tasks), yet highlighting this context is important for clarity. In terms of experiments, one limitation is that the authors restricted comparisons to real-time models, excluding stronger but slower SR networks (like SwinIR, Real-ESRGAN) from quantitative evaluation. Their reasoning is valid, those models fall far short of real-time speeds, but it does slightly limit the perspective on the ultimate achievable quality. Including at least a reference comparison to a top-performing offline model (even if only in an appendix or figure) could help quantify the quality gap between real-time and non-real-time SR. Another minor clarity point is the lack of detail in the main paper about the user study protocol (though presumably in the supplement): understanding how pairs were presented or how scores were aggregated would help gauge reliability. Finally, the work focuses on single-frame SR; there is no exploration of multi-frame/video SR algorithms that exploit temporal information. While this choice is appropriate for real-time efficiency, it’s an implicit limitation, methods that use neighboring frames (even in a limited way) might recover details or reduce flicker that single-frame methods cannot. Clarifying this as a conscious scope choice (and perhaps discussing the potential of efficient multi-frame SR in the future) would strengthen the methodology section. In summary, the paper could be more explicit about these limitations and choices, to enhance transparency and guide future extensions.

**Questions:**

See the weakness.

---

> ### Author Response · Authors · 2025-11-24
> **Part 1**
>
> Thank you for the detailed assessment and constructive suggestions. Below we address each point raised.
>
> > The dataset and experiments focus solely on YouTube videos and the prevalent codecs used there (VP9, H.264, AV1). This covers a large portion of online content, but it may not generalize to other domains or compression formats. For example, professional streaming or broadcast uses codecs like HEVC or VVC; these might produce different artifacts, and it’s unclear if models fine-tuned on StreamSR would handle them well.
>
> > Similarly, the resolution range is up to 1440p, the benchmark does not evaluate 1080p→4K (2160p) upscaling, a common high-end scenario.
>
> Our dataset focuses on VP9/H.264/AV1 because these dominate today’s streaming platforms, and because Creative Commons videos with aligned 360p-720p-1440p triplets are most reliably available on YouTube. We agree that broader codec coverage is valuable. As external validation, we evaluated EfRLFN on the open set of the Super-Resolution for Video Compression benchmark [1], which includes the H.264, HEVC, VVC, AV1, and AWS3 codecs. Although EfRLFN is designed for real-time operation, it produced stable visual outcomes without compression-specific artifacts and behaved consistently across codecs. It achieved the following results:
> * PSNR: 27.034
> * LPIPS: 0.201
> * SSIM: 0.952,
> * MDTVSFA: 0.579
>
> While EfRLFN falls behind large non-real-time models on objective metrics, these results indicate promising robustness outside the YouTube distribution.
>
> Regarding higher-end 1080p→4K upscaling: we view this as an important next step. Within CC-licensed YouTube content, 2160p videos with matching multi-resolution streams are significantly less common (less than 5% of all videos published by popular channels [2], and even less overall), which prevented collection of sufficiently many aligned pairs. Extending StreamSR toward 4K triplets is part of our planned future expansion.
>
> > The authors do show that models trained on StreamSR generalize to standard SR test sets, which is encouraging. However, the reverse direction (generalizing to content outside the YouTube distribution) remains uncertain.
>
> In the reverse-generalization direction, models fine-tuned on StreamSR perform competitively on standard benchmarks (Table 2, right), showing that despite domain specificity, the learned representations are not overfitted to YouTube-only statistics.
>
> > For instance, an ablation on the composite loss would be useful to see how each component (Charbonnier, VGG perceptual, Sobel) contributes to final quality, the paper mentions this composite greatly helped training stability, so showing a breakdown of its impact would solidify that claim.
>
> Figure 6(a) directly compares Charbonnier-only, VGG-only, Sobel-only, their pairwise combinations, and the full composite loss. Across training, the composite consistently yields the highest SSIM (0.01-0.03 above individual components), indicating complementary contributions: Charbonnier improves structural fidelity, VGG enhances perceptual alignment, and Sobel sharpens edges. The contrastive loss used in the original RLFN model also appears in the comparison; our loss shows faster and more stable convergence.
>
> > Similarly, since EfRLFN is built upon RLFN, it would be enlightening to have a direct comparison of training with vs. without the new StreamSR data for that model: e.g. how much does fine-tuning RLFN on StreamSR improve its performance on compressed videos (the authors do include this result) and on standard benchmarks (they report gains). Such comparisons are partly given (Table 2 and discussion), but a more detailed ablation could isolate the effect of the dataset itself on a model’s performance.
>
> Table 2 (left) isolates the effect of fine-tuning on StreamSR:
> * RLFN: Subjective score: 2.17 → 2.69;
> * SPAN: Subjective score: 2.55 → 3.13;
> * ESPCN: Subjective score: 2.48 → 2.49.
>
> Appendix Tables 9-11 extend this to 4×. On the 4× (360p→1440p) track, EfRLFN again achieves the best real-time performance among evaluated models. For example, it improves LPIPS over RLFN by 0.01–0.02 and raises SSIM consistently across datasets, while retaining real-time inference. These comparisons directly quantify dataset contribution without architectural changes.
>
>
> **References:**
>
> [1] https://videoprocessing.ai/benchmarks/super-resolution-for-video-compression.html
>
> [2] Li, Feng, Jae Won Chung, and Mark Claypool. "Three-year Trends in YouTube Video Content and Encoding." in SIGMAP, 2021.

---

> ### Author Response · Authors · 2025-11-24
> **Part 2**
>
> > While most such methods are too slow for real-time, it would be interesting to see if, for example, a two-frame fusion approach could outperform single-frame EfRLFN at the cost of some speed, this could inspire future research on reaching a better quality/speed middle ground.
>
> We agree that exploring efficient two-frame or multi-frame fusion is a compelling direction. Our focus on single-frame SR stems from the strict real-time setting: state-of-the-art multi-frame SR systems (e.g., BasicVSR++, RVRT) operate at 7-15 FPS on 720p inputs (Table 2), far below the 30 FPS requirement for real-time streaming. Nonetheless, we view hybrid designs that leverage minimal temporal context while maintaining real-time speed as promising future work.
>
>
> > The novelty of the EfRLFN model might be seen as somewhat incremental, it is essentially an improved variant of RLFN with known techniques (ECA attention, tanh activation) adopted from prior works. The paper does justify these choices well and supports them with ablations, but some readers may feel the model innovation is an optimization of existing components rather than a fundamentally new architecture. This is not a severe issue (pragmatic improvements are valuable for real-time tasks), yet highlighting this context is important for clarity.
>
> EfRLFN is intentionally lightweight, targeting practical real-time deployment. Although it builds upon RLFN, the ablation in Table 3 shows two architectural decisions: odd-symmetric tanh activations and lightweight ECA. While those techniques are not fundamentally new, they are usually underrepresented in real-time SR literature. As demonstrated, our use of tanh provides a gain over SPAN's shifted sigmoid, and our composite loss outperforms the one used in RLFN. In the revised paper we updated the Introduction to focus on the importance of real-time optimization, rather than presenting the novel architecture.
>
> > Including at least a reference comparison to a top-performing offline model (even if only in an appendix or figure) could help quantify the quality gap between real-time and non-real-time SR.
>
> > Another minor clarity point is the lack of detail in the main paper about the user study protocol (though presumably in the supplement): understanding how pairs were presented or how scores were aggregated would help gauge reliability.
>
> Table 2 and Appendix Table 11 include reference comparisons with Real-ESRGAN, SwinIR, BasicVSR++, and RVRT, providing an upper-bound comparison against offline models.
>
> Regarding the user study: Section 13 describes the protocol (30 ratings per pair, two verification questions per participant, 3,822 unique participants), and Figure 8 shows the user interface of the Subjectify platform. We moved a brief summary of these details into the main text for clarity.
>
>
>
> We appreciate the detailed feedback. We believe the above clarifications directly address the raised concerns and strengthen methodological transparency. We hope this resolves the remaining points and supports a positive reassessment of our submission.

---

### Author Response · Authors · 2025-11-28
**Rebuttal Summary**

Due to the reset of the discussion phase, we would like to summarize the rebuttal progress and the revisions made to our paper. We would also like to highlight that during the discussion stage, we thoroughly addressed reviewers’ concerns, which resulted in all “borderline reject” scores (4) being revised to “borderline accept” (6):

**yoXt: 4 -> 6**

**2ofZ: 4 -> 6**

**LP9y: 4 -> 6**

**9P2C: 6 -> 6, No response**

The reviewers raised the following concerns:

> Insufficient evidence that EfRLFN is “hardware-friendly” (**2ofZ**)

**Revision:** As suggested, we converted our model and baselines to ONNX and reported TensorRT runtime measurements to support our efficiency claims.

> Insufficient clarification regarding the absence of a temporal component (**2ofZ, LP9y**)

**Revision:** We explained the computational overhead introduced by temporal aggregation and why this makes it unsuitable for real-time super-resolution.

> Concerns about the positioning and contribution of the paper (**2ofZ, LP9y, yoXt**)

**Revision:** We emphasized the role of EfRLFN’s design in achieving the reported gains and refocused the Introduction to highlight the contribution of the StreamSR dataset.

>  Perceived insufficient novelty of the proposed model (**9P2C, 2ofZ, LP9y, yoXt**)

**Revision:** We detailed the importance of each EfRLFN component and clarified the architectural decisions in the revised manuscript.

> Lack of StreamSR dataset visualizations (**yoXt**)

**Revision:** We added representative dataset samples and highlighted the public availability of the dataset.

---

### Meta-Review · Area_Chair_56UT · 2026-01-02

**Summary:**

This submission presents "StreamSR," a comprehensive dataset of YouTube videos for benchmarking real-time super-resolution on compressed streaming content, and proposes "EfRLFN," an efficient real-time SR model. The paper addresses a gap in existing SR research by focusing on real-world compressed video artifacts rather than synthetic degradation. The reviewers raised several key concerns that informed the initial scores: insufficient evidence for hardware-friendly claims, limited architectural novelty of EfRLFN, concerns about the disconnect between video streaming problems and single-image SR solutions, lack of dataset visualizations, and questions about generalization beyond YouTube content.

**Reviewer Concerns:**

The authors' rebuttal effectively addressed most reviewer concerns. For reviewer's concern about insufficient hardware-friendly evidence, the authors provided ONNX and TensorRT runtime measurements showing EfRLFN's deployment efficiency. Regarding temporal component concerns, the authors clarified that real-time constraints necessitate single-image SR and demonstrated through user studies that EfRLFN outperforms NVIDIA VSR despite being frame-based. For positioning and contribution concerns raised by multiple reviewers, the authors reframed the paper to emphasize StreamSR as the primary contribution with EfRLFN as a strong baseline. Novelty concerns were addressed by detailing EfRLFN's specific architectural choices and their impact on compressed video restoration.

One concern that remains partially addressed is the generalization beyond YouTube content and codecs. While the authors provided some external validation on the Super-Resolution for Video Compression benchmark, showing consistent performance across codecs, the dataset's focus on YouTube videos remains a limitation.

**Reviewer Scores:**

Based on the discussion and rebuttal responses,  most reviewers were inclined to accept the manuscript. The paper makes a valuable contribution through the StreamSR dataset and comprehensive benchmarking, which will benefit the research community. Given that most  initially skeptical reviewers were satisfied with the rebuttal and indicated willingness to raise their scores, the AC recommend Accept.

---

### Decision · Program_Chairs · 2026-01-26

Accept (Poster)